# Adaptor protein supersaturation drives innate immune signaling and cell fate

**Alejandro Rodriguez Gama[1], Tayla Miller[1], Shriram Venkatesan[1], Jeffrey J Lange[1], Jianzheng Wu[1], Xiaoqing Song[1], William D Bradford[1], Malcolm Cook[1], Jay R Unruh[1], Randal Halfmann[1,2]***

[1]Stowers Institute for Medical Research, Kansas City, United States; [2]Department of Biochemistry and Molecular Biology, University of Kansas Medical Center, Kansas City, United States

## eLife Assessment

This **important** study investigates the self-assembly activity of all 109 human death-fold domains. The data collected using advanced microscopy and distributed amphifluoric FRET-based flow cytometry methods are **compelling** to support the "phase change battery" model that explains how signal amplification can occur without ATP consumption. This paper provides new insight into the thermodynamic control of protein phase behaviors within cells and will be of interest to those studying a variety of biological pathways involved in inflammatory responses and various forms of cell death.

***For correspondence:**
rhn@stowers.org

**Abstract** How minute pathogenic signals trigger decisive immune responses is a fundamental question in biology. Classical signaling often relies on ATP-driven enzymatic cascades, but innate immunity frequently employs death fold domain (DFD) self-assembly. The energetic basis of this assembly is unknown. Here, we show that specific DFDs function as energy reservoirs through metastable supersaturation. Characterizing all 109 human DFDs, we identified sequence-encoded nucleation barriers specifically in the central adaptors of inflammatory signalosomes, allowing them to accumulate far above their saturation concentration while remaining soluble and poised for activation. We demonstrate that the inflammasome adaptor ASC is constitutively supersaturated in vivo, retaining energy that powers on-demand cell death. Swapping a non-supersaturable DFD in the apoptosome with a supersaturable one sensitized cells to sublethal stimuli. Mapping all DFD nucleating interactions revealed that supersaturated adaptors are triggered to polymerize specifically by other DFDs in their respective pathways, limiting potentially deleterious crosstalk. Across human cell types, adaptor supersaturation strongly correlates with cell turnover, implicating this thermodynamic principle in the trade-off between immunity and longevity. Profiling homologues from fish and sponge, we find nucleation barriers to be conserved across metazoa. These findings reveal DFD adaptors as biological phase change materials in thermal batteries to power cellular life-or-death decisions on demand.

## Introduction

Innate immune signaling transduces small signals to robust responses such as programmed cell death and/or inflammation. Signaling occurs when danger- or pathogen-associated molecular patterns (D/PAMPs) activate cognate receptor proteins that then activate effector proteins such as caspases, typically via one or more intermediary proteins known as adaptors.

While the binding of a D/PAMP to receptors initiates signaling, this initial interaction releases insufficient energy to directly change cell state. That tiny signal must be amplified in an energy-consuming process that is fundamental to understanding the architecture and evolution of these critical signaling networks (*Mehta et al., 2016*; *Bérut et al., 2012*; *Goldbeter and Koshland, 1981*). Innate immune signaling networks from bacteria to humans amplify signaling through interactions between nonenzymatic death fold domains (DFDs), which comprise the caspase recruitment domain (CARD), death domain (DD), death effector domain (DED), and pyrin domain (PYD) subfamilies (*Kagan et al., 2014*; *Aravind et al., 2024*; *Wu et al., 2025*; *Wu, 2013*; *Kobe et al., 2025*). However, the energetic basis for DFD function has been the subject of considerable speculation but remains largely unresolved (*Figure 1A*).

DFDs commonly form paracrystalline polymers that can template their own growth when free subunits exceed their saturation concentration ($C_{sat}$). Polymers are functionally initiated by D/PAMP-bound receptor oligomerization. To preclude signaling through spontaneous nucleation, DFDs are presumed to be effectively subsaturated prior to activation, through a combination of low basal expression, subcellular compartmentalization, post-translational modifications, and regulatory interactions (*Kagan et al., 2014*; *Huoh and Hur, 2022*; *Seyrek et al., 2020*; *Zheng et al., 2020*). We previously discovered, however, that the DFD-containing adaptor Bcl10 exhibits an intrinsic (sequence-encoded) nucleation barrier that is large enough to support persistent deep supersaturation in vivo, allowing it to amplify signaling independently of orthogonal input (*Rodriguez Gama et al., 2022*).

The increased ordering of DFD subunits in polymers relative to those in solution implies that polymerization releases heat. Here, we propose that DFD-containing signalosomes broadly function as a form of thermal battery – an on-demand energy reservoir that can be discharged via latent heat release through a specific external input (*Sarbu and Sebarchievici, 2018*). Energy storage is accomplished via metastable supersaturation by a phase change material. A familiar example of a phase change material is the sodium acetate solution inside reusable hand warmers, which releases heat when the metal disk inside is pressed to nucleate crystalline sodium acetate trihydrate. Supersaturation is a state where a solute's concentration exceeds $C_{sat}$ yet remains in solution due to a structurally determined nucleation barrier. In the context of innate immune signalosomes, at least one DFD would function as the phase change material by constitutively exceeding its $C_{sat}$, allowing the signalosome to assemble immediately upon D/PAMP-triggered nucleation. Although heat will necessarily be released during the phase transition, it dissipates far too quickly to influence downstream signaling (*Song et al., 2021*). This mechanism would allow cells to respond to D/PAMPs decisively and independently of metabolism, which is frequently compromised during infection (*Thaker et al., 2019*). For DFDs to function in this manner, their endogenous concentration would need to greatly exceed their respective $C_{sat}$ values, while they nevertheless remain soluble over timescales spanning the window of vulnerability to infection, that is the full lifetimes of cells. Cells would therefore be continuously susceptible to spontaneous death and inflammation through stochastic nucleation events, imposing a fundamental tradeoff between immunity and longevity.

We here use a combination of biophysical, bioinformatic, and cytological approaches to investigate the capacity and functional relevance of supersaturation by human DFDs. Our results collectively uncover the energetic basis for signal amplification by DFDs and in turn a thermodynamic drive to die.

## Results

### A select subset of DFDs has intrinsic nucleation barriers enabling persistent supersaturation

To systematically survey the ability of DFDs to supersaturate, we compiled an exhaustive set of 109 structurally independent human DFDs (*Supplementary file 1* and *Figure 1—figure supplement 1A–B*) and then characterized their tendency to spontaneously self-assemble in near-physiological conditions while minimizing interference from other proteins. For this purpose, we used distributed amphifluoric FRET (DAmFRET) in an orthogonal eukaryotic host that completely lacks endogenous DFDs and their associated regulatory machinery – *Saccharomyces cerevisiae*. DAmFRET produces a snapshot of the population-level distribution of single-cell measurements of ratiometric FRET ('AmFRET') between two fluorescent forms of the same protein species (*Khan et al., 2018*). The data revealed a diversity of behaviors (*Figure 1B–C*, *Figure 1—figure supplement 1C–D*, and *Supplementary file 1*), ranging

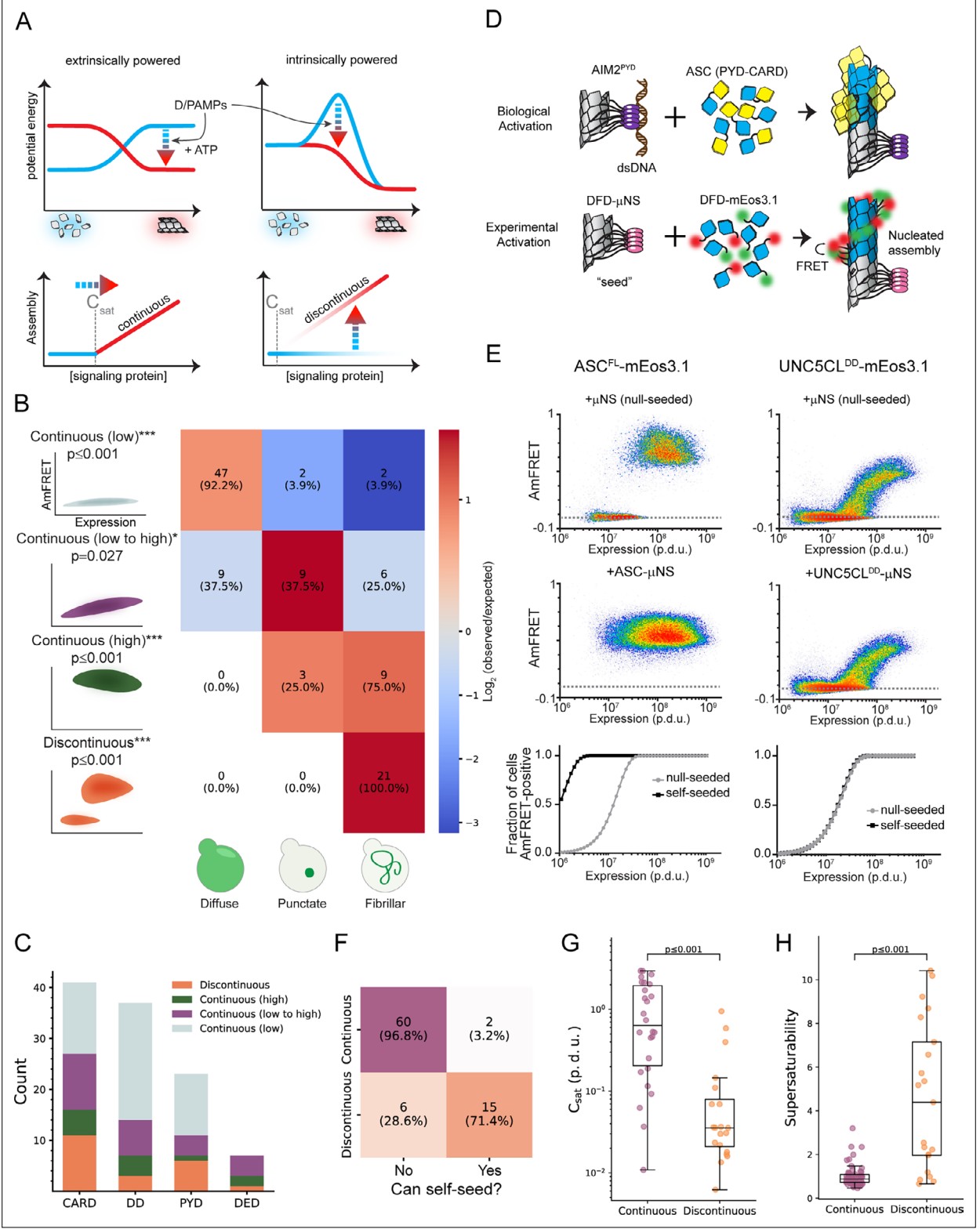

**Figure 1.** A select subset of DFDs has intrinsic nucleation barriers enabling persistent supersaturation. (**A**) Schematic diagram illustrating two models for signal amplification through protein self-assembly. **Top left:** Extrinsic model, where D/PAMP-binding coupled with nucleotide hydrolysis stabilizes active assemblies (red glow) relative to solute precursors (blue glow). This model is exemplified by localized actin polymerization downstream of many cell surface receptors (*Padrick and Rosen, 2010*), but could also occur indirectly by, for example, phosphorylation-mediated release of solubilizing factors. **Bottom left:** DFDs that function in this way will assemble promptly and monotonously above their saturation concentration ($C_{sat}$). **Top right:**

*Figure 1 continued on next page*

*Figure 1 continued*

Intrinsic model, where the protein is supersaturated at rest but prevented from assembling by a sequence-encoded nucleation barrier. D/PAMP-binding eliminates the barrier, releasing the energy of supersaturation to drive assembly. The models are not mutually exclusive. **Bottom right:** DFDs that function in this way will remain soluble above $C_{sat}$ until stochastic nucleation, creating a discontinuous relationship of assembly to concentration across a population of cells. (**B**) Illustration showing how the concentration dependence of self-assembly as classified by DAmFRET relates to the subcellular morphology of self-assemblies classified by high-throughput confocal microscopy. 'Continuous' and 'discontinuous' classifications describe the relationship of self-assembly (AmFRET, y-axis) to expression level (x-axis) for each DFD. Discontinuous DFDs exhibit a range of concentrations where self-assembly occurs stochastically, indicating an intrinsic nucleation barrier. The four instances of visible assemblies despite no AmFRET-positive cells are presumed to result from those DFDs partitioning with other cellular components or endogenous condensates wherein they remain too dilute to FRET. Cells in the matrix are colored according to the log2 ratio of observed to expected frequencies. The indicated p-values (adjusted for multiple hypotheses with Bonferroni correction) were obtained with an exact multinomial test using the total frequencies of each morphology (diffuse = 0.52, punctate = 0.13, fibrillar = 0.35). The sample sizes and expected frequencies (diffuse, punctate, fibrillar) are $n_{continuous\ (low)}$ = 51 (26.4, 6.6, 17.9), $n_{continuous\ (low\ to\ high)}$ = 24 (12.4, 3.1, 8.4), $n_{continuous\ (high)}$ = 12 (6.2, 1.6, 4.2), $n_{discontinuous}$ = 21 (10.9, 2.7, 7.4). (**C**) Distribution of DAmFRET classifications across the four subfamilies of DFDs. (**D**) Schematic diagram of our experimental design to assess the ability of each DFD to seed itself. Top: Biological activation of an exemplary signalosome – the AIM2 inflammasome – occurs when the receptor AIM2 oligomerizes on the multivalent PAMP, dsDNA, and then templates the assembly of the adaptor protein, ASC. Bottom: Experimental paradigm to test for supersaturation mimics biological activation, by expressing each DFD in trans with the same DFD expressed as a fusion to µNS, a modular self-condensing protein. AmFRET-positivity will only occur if the µNS fusion templates subsequent self-assembly by the non µNS-fused DFD. (**E**) Representative DAmFRET data contrasting two self-assembling DFDs – one that is supersaturable (left) and the other that is not (right). The plot for the supersaturated protein exhibits a discontinuous distribution of AmFRET across the expression range (top and bottom). The discontinuity is eliminated, with all cells moving to the AmFRET-positive population, by expressing the protein in the presence of genetically encoded seeds (middle). The dashed horizontal lines approximate the mean AmFRET value for monomeric mEos. Procedure defined units (p.d.u.). (**F**) Contingency table showing that discontinuous DFDs tend to be self-seedable. Each DFD was co-expressed with an orthogonally fluorescent µNS-fused version of the same DFD. Fisher's exact test revealed an association between continuity and self-seedability, $n_{continuous}$ = 62, $n_{discontinuous}$ = 21 (p<0.001). (**G**) Boxplot comparing the $C_{sat}$ values (as approximated by $C50_{seeded}$) of continuous and discontinuous DFDs. Discontinuous DFDs have significantly lower $C_{sat}$, indicating greater stability of the assemblies. Mann-Whitney U=457, $n_{continuous}$ = 26, $n_{discontinuous}$ = 20 (p<0.001). (**H**) Boxplot comparing supersaturability, represented as the fold change reduction in C50 by seeding ($C50_{stochastic}$ - $C50_{seeded}$), of continuous and discontinuous DFDs. The C50 values were more strongly reduced by seeding for discontinuous DFDs than for continuous DFDs. Mann-Whitney U=164, $n_{continuous}$ = 58, $n_{discontinuous}$ = 21 (p<0.001).

The online version of this article includes the following figure supplement(s) for figure 1:

**Figure supplement 1.** Sequence, imaging, and DAmFRET analysis reveal diverse sequence-encoded phase behaviors of DFDs.

**Figure supplement 2.** Self-assembly involves subunit interfaces shared with solved DFD polymer structures.

from no self-association to self-association in all cells at all concentrations. Importantly, the results closely agree with observations of individual DFDs in the literature. Considering the adaptor FADD as an example, its DD was monomeric, whereas its DED assembled robustly, consistent with prior observations that the DD forms exclusively hetero-oligomers while the DED forms homopolymers (*Jang et al., 2014*; *Fosuah et al., 2025*; *Wang et al., 2010*). Twenty-one DFDs transitioned from no to high AmFRET in a discontinuous manner, the signature of a large intrinsic nucleation barrier (*Rodriguez Gama et al., 2022*; *Khan et al., 2018*; *Kandola et al., 2023*).

Nucleation barriers increase with the entropic cost of assembly (*Khan et al., 2018*; *Buell, 2017*). Consequently, assemblies with large barriers tend to be more ordered than those without. Ordered assembly by DFDs occurs as a two-dimensional array twisted into a one-dimensional polymer (*Lin et al., 2010*; *Lu et al., 2014*; *Rodríguez Gama et al., 2021*) that can manifest as microscopically visible filaments in cells (*Rodriguez Gama et al., 2022*; *Lu et al., 2014*; *Shearwin-Whyatt et al., 2000*). To evaluate the expected relationship between nucleation barriers and ordered assembly, we used high-throughput confocal microscopy to examine the subcellular distribution of each protein. We observed subcellular assemblies for most of the DFDs that populated AmFRET-positive states, but not those with entirely low AmFRET (*Figure 1B* and *Figure 1—figure supplement 1E*). As predicted, the assemblies of DFDs that had transitioned discontinuously had fibrillar morphologies, whereas those that had transitioned continuously (low to high DAmFRET) instead mostly formed spherical or amorphous puncta (*Figure 1B*, *Figure 1—figure supplement 1F–H*).

For discontinuous DFDs, the soluble protomers in the low FRET cells are hypothetically poised to assemble; they just lack a structural template to get them started. We tested this hypothesis by co-expressing genetically encoded 'seeds' (*Figure 1D–E*; *Rodriguez Gama et al., 2022*) and found that the seeds caused cells to switch to the high AmFRET population for most discontinuous DFDs but not continuous DFDs (*Figure 1F*, *Figure 1—figure supplement 1I*, *Supplementary file 1*). We further

demonstrated that assembly is more consistent with native DFD subunit interfaces than amyloid-like misfolding (Supplemental Information, *Figure 1—figure supplement 2A–B*). This distinction is important because amyloid is the only other form of assembly that has been experimentally shown to be capable of forming a discontinuous DAmFRET profile (*Khan et al., 2018*; *Posey et al., 2021*). Altogether, these data confirm that a subset of DFDs can supersaturate in a soluble form to power subsequent switch-like assembly.

The magnitude of supersaturation in vivo is determined by the ratio of a protein's total concentration to its $C_{sat}$, which reflects the strength of interactions between subunits in its assembled structure (*Kar et al., 2022*). Total concentration and $C_{sat}$ can evolve independently by gene regulation and DFD sequence, respectively. Consequently, if the function of discontinuous DFDs involves supersaturation, we would expect evolution to have lowered their $C_{sat}$ while raising their expression relative to continuous DFDs. To test these predictions, we first analyzed the relationship of nucleation barriers to $C_{sat}$, as determined by each protein's transition concentration in the presence of seed. As expected, discontinuous DFDs exhibited lower $C_{sat}$ values than continuous DFDs (*Figure 1G*, *Supplementary file 1*), and in the absence of seed achieved soluble concentrations that exceeded them by at least fourfold on average (*Figure 1H*).

## Nucleation barriers are restricted to innate immune signalosome adaptors

We next evaluated the relationship of nucleation barriers to DFD concentrations in vivo. Using published proteomic datasets (*Wang et al., 2015*; *Karlsson et al., 2021*; *Jiang et al., 2020*), we found that proteins with discontinuous DFDs tend to be expressed more abundantly than those with continuous DFDs, both at the cell type and tissue levels (*Figure 2A*, *Figure 2—figure supplement 1A–B* and *Supplementary file 1*). We then used a high-coverage transcriptomics dataset (*Uhlen et al., 2019*) to ask how $C_{sat}$ values relate to DFD gene expression in primary immune cells. Remarkably, transcript abundances significantly anticorrelated with $C_{sat}$ in 17 of the 18 canonical immune cell populations (*Figure 2B* and *Figure 2—figure supplement 1C*). Such a relationship is highly unusual for soluble proteins, whose expression instead tends to strongly correlate with $C_{sat}$ (*Vecchi et al., 2020*). These analyses collectively indicate that discontinuous DFDs are likely to be functionally supersaturated in their endogenous physiological contexts.

Given that nucleation barriers are restricted to only a subset of DFDs, we asked if the DFDs with nucleation barriers tend to have different signaling roles than those without. From our prior demonstrations of prion-like activity by ASC, MAVS, and BCL10 (*Rodriguez Gama et al., 2022*; *Cai et al., 2014*), we suspected an enrichment among adaptors. However, DFD proteins have not been systematically evaluated with respect to adaptor function. We therefore focused on the defining property of 'adaptors' physically connecting other proteins by quantifying their centrality in the DFD protein-protein interaction network. Proteins with high 'degree centrality' are hub-like, with many direct interactions. Proteins with high 'betweenness centrality' lie on the shortest paths between other nodes and therefore act as bottlenecks in the network. Assessing both measures (see Methods), we observed that discontinuous and/or seedable DFD proteins have significantly greater degree and betweenness centralities (*Figure 2C*, *Figure 2—figure supplement 1D-E*). As a point of comparison, we performed the same analysis for the domain family most closely resembling DFDs – Sterile Alpha Motif (SAM). SAM domains have similar size, structure, number (76 in humans), and function as DFDs, often serving as polymeric scaffolds of signaling complexes (*Bienz, 2020*; *Qiao and Bowie, 2005*; *Knight et al., 2011*). Unlike DFDs, however, SAM domains cannot supersaturate (*Rodríguez Gama et al., 2021*). Consistent with that, we found that they also have much lower betweenness centrality in their physical interaction network as compared to discontinuous DFDs (p=0. 0003; *Figure 2—figure supplement 1F*).

For a DFD nucleation barrier to enable on-demand assembly, the DFD must remain supersaturated even as a full-length (FL) protein. This means that other parts of the protein must not trigger nucleation prematurely, as through homo-oligomerization. Nor should they disfavor the assembled state, which would raise $C_{sat}$. We therefore evaluated the phase behaviors of 21 diverse DFD-containing FL multi-domain proteins. Most of the proteins behaved the same way as their respective DFDs (*Figure 2D*, *Figure 2—figure supplement 2A*, and *Supplementary file 2*). Six proteins suppressed the ability of their DFDs to supersaturate, resulting in continuous low or moderate DAmFRET, suggesting they

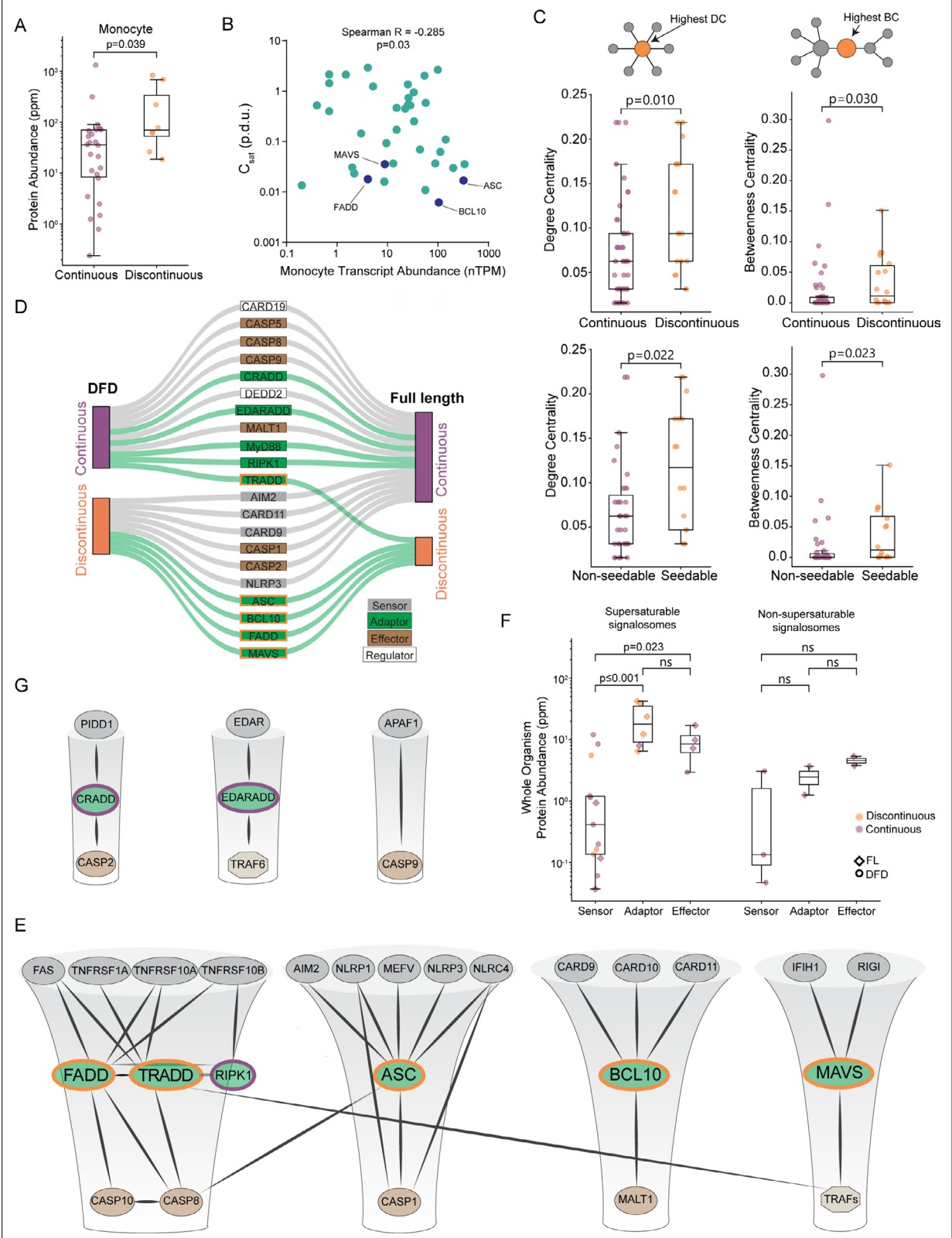

**Figure 2.** Nucleation barriers are a characteristic feature of inflammatory signalosome adaptors. (**A**) Boxplot of DFD-containing protein abundances in monocytes, showing that discontinuous DFDs have higher endogenous expression levels. Mann-Whitney U=53, $n_{continuous}$ = 26, $n_{discontinuous}$ = 8 (p=0.039). Protein abundance values are from PAXdb (**Wang et al., 2015**). (**B**) Scatter plot of DFD gene expression in monocytes (normalized transcripts per million) and $C_{sat}$ values. Spearman $R$=−0.285 (p=0.03). Adaptor DFDs are labeled. Dataset obtained from the Human Protein Atlas. (**C**) Top: box plots of

*Figure 2 continued*

degree centrality (left) and betweenness centrality (right) of continuous and discontinuous DFDs in the endogenous network of physically interacting DFD proteins, showing that the latter are more centrally positioned. Degree centrality Mann-Whitney U=242.0 (p=0.010); betweenness centrality Mann-Whitney U=274.0 (p=0.030); $n_{continuous}$ = 46, $n_{discontinuous}$ = 18. Bottom: box plots of centrality measures of non-seedable and seedable DFDs, showing that the latter are more centrally positioned. Degree centrality Mann-Whitney U=167.5 (p=0.022); betweenness centrality Mann-Whitney U=172.5 (P=0.023); $n_{non-seedable}$ = 35, $n_{seedable}$ = 16. (**D**) Visualization of how the DAmFRET profiles of isolated DFD domains (left) change in their full-length contexts (right), showing that only adaptor proteins (green connections) tend to retain discontinuous transitions in their full-length context. (**E**) Subnetworks of prominent signalosome adaptor proteins that were found to be supersaturable. Edges connect nodes with experimentally determined physical interactions with confidence >0.9 in STRING. All proteins shown have DFDs except TRAFs. Each adaptor's node size is proportional to its supersaturability score. (**F**) Comparison of protein abundances at the whole body level for the signalosome components in **E** (left) and **G** (right), showing that adaptors are more highly expressed for the former. Protein abundance values are from PAXdb (*Wang et al., 2015*). p-Values are from Mann-Whitney test. For supersaturable signalosomes: $n_{sensor}$ = 13, $n_{adaptor}$ = 6, $n_{effector}$ = 4; sensors and adaptors, U=4.0 (p<0.001); sensors and effectors, U=6.0 (p=0.023); adaptors and effectors, U=18.0 (p=0.257). For non-supersaturable signalosomes: $n_{sensor}$ = 3, $n_{adaptor}$ = 2, $n_{effector}$ = 2; sensors and adaptors, U=1.0 (p=0.400); sensors and effectors, U=0.0 (p=0.200); adaptors and effectors, U=0.0 (p=0.333). (**G**) Subnetworks of signalosomes lacking supersaturable DFDs. Edges connect nodes with experimentally determined physical interactions with confidence >0.9 in STRING. All proteins shown have DFDs except TRAF6.

The online version of this article includes the following figure supplement(s) for figure 2:

**Figure supplement 1.** Proteins with DFDs that have seedable and/or discontinuous DAmFRET are central to their physical interaction networks are more likely to be supersaturated in vivo.

**Figure supplement 2.** Proteins characterized as signaling adaptors display discontinuity in their DFD and FL context.

**Figure supplement 3.** Full-length proteins displaying discontinuous profiles can be self-seeded.

form non-nucleated oligomers. To determine if the DFDs are in an active polymer configuration within these oligomers, we tested the ability of FL NLRP3 to nucleate its cognate adaptor, ASC. It failed to do so (*Figure 2—figure supplement 2B*), revealing that the oligomers are 'autoinhibited' as has been previously demonstrated for multiple DFD-containing receptors and effectors (*Huoh and Hur, 2022*; *Holliday et al., 2019*; *Ohto et al., 2022*; *Hochheiser et al., 2022*; *Sommer et al., 2005*; *Andreeva et al., 2021*; *Chuenchor et al., 2014*; *Green, 2022*). This implies that the DFD within FL NLRP3 (and presumably other DFDs that lost supersaturability in their FL context) cannot *self*-assemble; instead, its joining the polymer structure is driven by the energy released by ligand (e.g. D/PAMP) binding. This stoichiometric requirement limits spurious activation that would otherwise kill or inflame cells unnecessarily (*Huoh and Hur, 2022*; *Holliday et al., 2019*; *Ohto et al., 2022*; *Hochheiser et al., 2022*; *Sommer et al., 2005*; *Andreeva et al., 2021*; *Chuenchor et al., 2014*). In contrast to these autoinhibited proteins, five of the FL proteins instead retained or enhanced their DFDs' nucleation barriers: ASC, BCL10, FADD, MAVS, and TRADD (*Figure 2D*, *Figure 2—figure supplement 2A*, and *Supplementary file 2*). Testing a subset of FL proteins further, we found that only this latter group can be self-seeded (*Figure 2—figure supplement 3A*). These proteins collectively span all four DFD subfamilies, and all of them function as major adaptors in innate immune signalosomes (*Figure 2E*). ASC, BCL10, MAVS, and TRADD drive inflammation and/or inflammatory cell death downstream of D/PAMP sensing (*Martinon et al., 2002*; *Seth et al., 2005*; *Kawai et al., 2005*; *Micheau and Tschopp, 2003*; *Gross et al., 2006*; *Bertin et al., 2000*; *Bertin et al., 2001*). FADD signals downstream of death ligands and certain D/PAMPs through various signalosomes that can be either anti- or pro-inflammatory (*Henry and Martin, 2017*; *Tummers et al., 2020*; *Chen et al., 2005*; *Gurung et al., 2014*; *Mouasni and Tourneur, 2018*; *Balachandran et al., 2004*). In contrast, signalosomes with primarily non-immunity functions – the apoptosome, the PIDDosome, and the ectodysplasin (EDA) receptor complex (*Li et al., 1997*; *Tinel and Tschopp, 2004*; *Headon et al., 2001*) – lacked supersaturation (*Figure 2G*). Consistent with their functioning through supersaturation, only the innate immune set of adaptors is more abundantly expressed than their cognate receptors in the human body (*Figure 2F*). These data collectively identify the adaptors, specifically, of inflammatory signalosomes as energy reservoirs for signal amplification.

## Nucleation barriers may facilitate signal amplification in human cells

To explore the functionality of supersaturation, we focused on two modes of programmed cell death signaling that differed in this regard: intrinsic apoptosis and pyroptosis. The former occurs downstream of persistent intracellular stresses (*Gong et al., 2019*; *Nano et al., 2023*; *Häcker and Haimovici, 2023*). The latter is instead triggered by minute levels of D/PAMPs and therefore involves greater

signal amplification. Our DAmFRET analyses revealed that CARDs of the apoptosome lack nucleation barriers, whereas CARDs of the inflammasome, which drives pyroptosis, have nucleation barriers.

To test if the absence of nucleation barriers limits the sensitivity of the apoptosome, we adapted an optogenetic approach (*Rodriguez Gama et al., 2022*; *Park et al., 2017*; *Shkarina et al., 2022*; *Kennedy et al., 2010*) to precisely control the initiation of intrinsic apoptosis. HEK293T cells lack the inflammasome constituents NLRC4, ASC, and CASP1, allowing us to repurpose their DFDs in this cell line. Accordingly, we transduced the cells with mScarlet-I fusions of either the non-supersaturable WT apoptosome effector, CASP9, or a chimeric version that harbored the supersaturable DFD of CASP1 in place of its own (CASP9$^{CASP1CARD}$). We simultaneously transduced the cells with blue light-inducible seeds of the cognate upstream DFDs – opto-APAF1 or opto-NLRC4, respectively (*Figure 3A*). We then assessed signal propagation through the reconstituted pathways by measuring activation of CASP9's downstream target, CASP3/7, after 1 min of blue light stimulation. Both cell lines activated a fluorescent CASP3/7 reporter to the same extent (*Figure 3B*), confirming that the different modifications do not sterically interfere with the proteins' activities.

To now assess signal amplification, we evaluated both the persistence of DFD assemblies and their ability to commit cells to apoptosis following an otherwise sublethal stimulus. We anticipated that cells expressing non-supersaturable CASP9 will (1) form transient clusters and consequently (2) survive a short blue light stimulus. In contrast, cells expressing supersaturable CASP9 will (1) form perdurant clusters that signal indefinitely, and consequently (2) die even after a short blue light stimulus.

To test prediction 1, we stimulated the cells with blue light for 1 min and monitored subsequent protein localization. We found that both pairs of proteins formed puncta in essentially all cells within one minute (*Figure 3C–D*; *Figure 3—video 1*). The puncta with non-supersaturable CASP9 then dissolved over the course of the next 10 min. In contrast, the puncta with supersaturable CASP9 instead continued to grow at a constant rate for at least the next 20 min (*Figure 3C–D*; *Figure 3—video 2*), confirming that the protein's drive to polymerize exists even without the stimulus.

To test prediction 2, we measured cell death 2 hr after the initiation of blue light stimulation for either 1 min or the entire 2 hr. Neither pair of proteins induced cell death in the absence of blue light (*Figure 3E*). Non-supersaturable CASP9 induced death in approximately 18% and 55% of cells following the short and long stimulation, respectively (*Figure 3E*), confirming the expected dose-dependence of signaling. In contrast, supersaturable CASP9-induced death in most cells even upon short stimulation is consistent with the expected amplification of signaling due to supersaturation-mediated DFD polymerization. We note that the increased stability of NLRC4$^{CARD}$ and CASP1$^{CARD}$ polymers relative to APAF1$^{CARD}$ and CASP9$^{CARD}$ oligomers could increase signal amplification irrespective of polymerization; specifically, due to the slower dissolution of assemblies that formed even during stimulation which could provide CASP9$^{CASP1CARD}$ more time to activate more molecules of CASP3/7. Consequently, further work will be required to distinguish the contributions of each mechanism to amplification.

## Some innate immune adaptors are endogenously supersaturated

We next asked if pyroptosis and extrinsic apoptosis are indeed powered by adaptor supersaturation in vivo. We first induced pyroptosis in human THP-1 monocytes by treating them with poly(dA:dT), a ligand for the inflammasome receptor, AIM2. By 18 hr, approximately 70% of the cells were dead or dying. We then deleted *PYCARD* to determine if cell death depended on ASC, the adaptor for the inflammasome. Death was delayed, but not eliminated (*Figure 4A*). We then also deleted *FADD*, encoding an adaptor that has been shown to co-assemble with AIM2 in pathogen-triggered PANoptosis (*Lee et al., 2021*). Death was reduced further still (*Figure 4A* and *Figure 3—figure supplement 1B*), confirming that AIM2 activation can induce cell death through both ASC and FADD.

To eliminate the potentially confounding effects of orthogonal dsDNA sensors (*Maelfait et al., 2020*), we next placed AIM2$^{PYD}$ under blue light control and confirmed that the resulting fusion protein – 'opto-AIM2' – grants direct control over seed formation (*Figure 4B and C*, top). Using this system, a transient blue light exposure induced AIM2 clustering within seconds and cell death within 10 min (*Figure 4C–D* and *Figure 4—videos 1; 2*). Next, we evaluated if cell death depended on ASC or FADD. We found that stimulating WT cells with a 1-s blue laser pulse every 15 min killed essentially all of them within 1 hr (*Figure 4E*). In contrast, only half of cells lacking ASC (*PYCARD*-KO) died, and

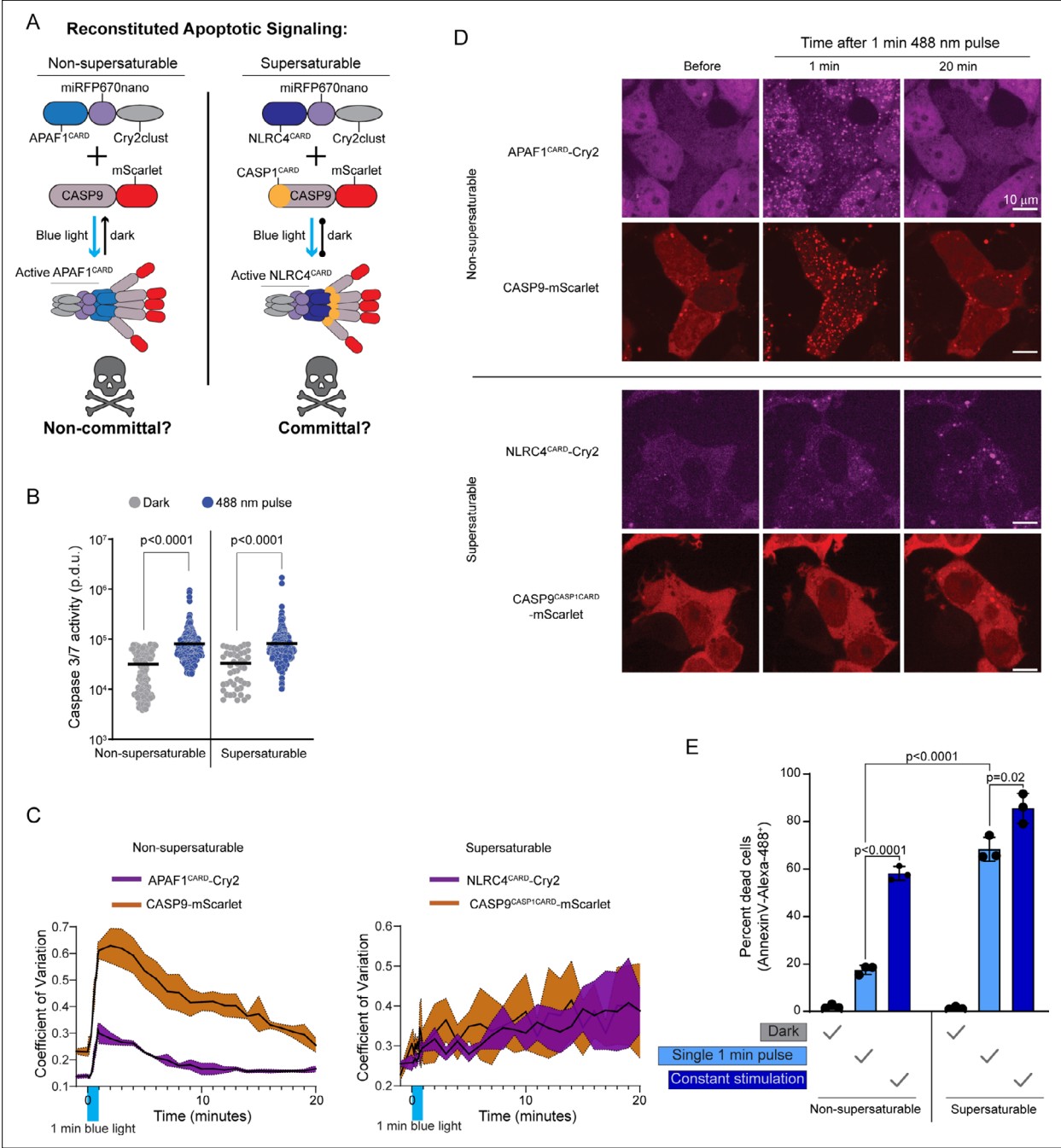

**Figure 3.** Nucleation barriers may facilitate signal amplification in human cells. (**A**) Schematic diagram of experiment in HEK293T cells to reconstitute the apoptosome with optogenetic control, in either a non-supersaturable or supersaturable format. The non-supersaturable format comprises CASP9 activated by APAF1$^{CARD}$ (as in the native apoptosome); the supersaturable format comprises chimeric CASP9 with CASP1$^{CARD}$ replacing CASP9$^{CARD}$ (CASP9$^{CASP1CARD}$), activated by chimeric APAF1 with NLRC4$^{CARD}$ in place of APAF1$^{CARD}$. Blue light triggered assembly in both cases, but subsequent continued assembly in the dark only occurred for the supersaturated format. (**B**) Caspase 3/7 activity reporter fluorescence intensities in the absence of stimulation or after 1 min of 488 nm stimulation for cell lines expressing the non-supersaturable or supersaturable pairs, showing that both pairs comparably activate caspase 3/7 while oligomerized. APAF1$^{CARD}$-Cry2 + CASP9-mScarlet-I, dark n=163, pulse n=375, Mann-Whitney U=11,362 (p<0.0001). NLRC4$^{CARD}$-Cry2 + CASP9$^{CASP1CARD}$, dark n=46, pulse n=305, Mann-Whitney U=4,253 (p<0.0001). (**C**) Coefficient of variation (CV) of fluorescence distribution in HEK293T cells expressing the indicated protein pairs after a single 1 min 488 nm laser activation. Top, APAF1$^{CARD}$-Cry2, and CASP9-mScarlet-I display rapid cluster formation that dissociates by 20 min. Bottom, NLRC4$^{CARD}$-Cry2 and chimeric CASP9$^{CASP1CARD}$ cluster less rapidly but the clusters continue to grow indefinitely. (**D**) Representative images from experiment in C. Clusters of APAF1$^{CARD}$-Cry2 and CASP9-mScarlet-I form then dissociate, while NLRC4$^{CARD}$-Cry2 and CASP9$^{CASP1CARD}$ clusters only get larger. Scale bar 10 μm. (**E**) Quantification of cell death of the HEK293T

*Figure 3 continued on next page*

*Figure 3 continued*

chimeric cells (as in A) using Annexin V-Alexa 488 staining, either 2 hr after a single 1 min pulse of 488 nm laser, or after 2 hr of 'constant' stimulation whereby cells were subjected to a 1 s pulse every 1 min. p-Values derived from t-test.

The online version of this article includes the following video and figure supplement(s) for figure 3:

**Figure supplement 1.** Characterization of engineered THP-1 cell lines and apoptosome assembly, and correlation of DFD supersaturation with cell mortality in the human body.

**Figure 3—video 1.** Time-lapse video of cells expressing opto-APAF1 (top) and CASP9-mScarlet-I (bottom) showing 1 min of blue light induction and subsequent recovery, related to *Figure 3*.

https://elifesciences.org/articles/107962/figures#fig3video1

**Figure 3—video 2.** Time-lapse video of cells expressing opto-NLRC4 (top) and CASP9(CASP1CARD)-mScarlet-I (bottom), showing 1 min of blue light induction and subsequent recovery, related to *Figure 3*.

https://elifesciences.org/articles/107962/figures#fig3video2

they did so with delayed kinetics consistent with FADD-driven signaling (*Tummers et al., 2020*; *Place et al., 2021*; *Roncaioli et al., 2023*).

Finally, we directly assessed inflammasome nucleation by AIM2PYD seeds. To do so, we reconstituted the *PYCARD*-KO with mScarlet-I-tagged ASC and titrated its expression to well below endogenous levels so as to circumvent any potential for the fusion tag to enhance the protein's oligomerization (although this would be unexpected; *Bindels et al., 2017*; *Figure 3—figure supplement 1C–E*). We then tracked AIM2PYD and ASC localization following a 10 laser pulse. Both proteins began to cluster almost immediately. AIM2PYD cluster growth nearly stopped by 3 min, consistent with excited Cry2 relaxation on a similar time scale (*Park et al., 2017*; *Che et al., 2015*). In contrast, ASC clusters continued rapid growth for more than 10 min (*Figure 4F*), demonstrating that their drive to do so is independent of the triggering stimulus. The plasma membrane concomitantly permeabilized as a result of gasdermin D (GSDMD) activation (*He et al., 2015*), and to an extent that increased with the level of ASC expression (*Figure 4G*). These data collectively confirm that endogenous ASC is highly supersaturated prior to stimulation, and that the extent of supersaturation determines the extent of signal amplification.

Given that inflammatory signalosomes frequently initiate programmed cell death, and that a kinetic barrier governs their activity, the susceptibility of cells to aberrant cell death through spontaneous nucleating fluctuations is expected to increase with the level of adaptor supersaturation in vivo. This would constrain a cell's life expectancy if such nucleation occurs with appreciable frequency. To explore this possibility, we asked how adaptor supersaturation (as approximated by the ratio of mRNA levels to $C_{sat}$ values) relates to the turnover rates of each cell type in the human body (*Sender and Milo, 2021*). We found that short-lived cells such as monocytes indeed have greater DFD supersaturation than long-lived cells such as neurons (*Figure 4H*). Among individual DFD proteins, cell turnover correlated especially strongly with the expression level of ASC (*Figure 3—figure supplement 1F–G*), suggesting that life expectancy may be limited by the thermodynamic drive for inflammatory signal amplification.

## The nucleating interactome is highly specific

DFDs share the same fold and are co-expressed in many of the same cells. While spontaneous nucleation of supersaturable DFDs poses an inherent risk of inadvertently activating end fate cell signaling, this possibility is multiplied by the risk of cross-activation between DFDs in different pathways. To what extent do they nucleate each other? To determine specificity and systematically map the nucleating interactome of DFDs, we mated our library of seed-expressing yeast strains with a library of strains expressing each mEos-fused DFD (*Figure 5—figure supplement 1A*) to create over 10,000 arrayed diploid yeast strains that we then screened by DAmFRET (*Figure 5—figure supplement 1B and C*).

In total, we identified 171 nucleating interactions, representing just ~1.6% of the total library (*Figure 5—figure supplement 1D–F* and *Supplementary file 4*). The interactions were largely constrained not just to members of the same subfamily, but to members of the same signaling subnetwork even within subfamilies (*Figure 5A–B*). For example, CARDs of the CBM signalosome nucleated each other but not CARDs of the inflammasome, and vice versa. As an exception, PYDs of the inflammasome and DEDs of the Death-Inducing Signaling Complex (DISC) nucleated each other

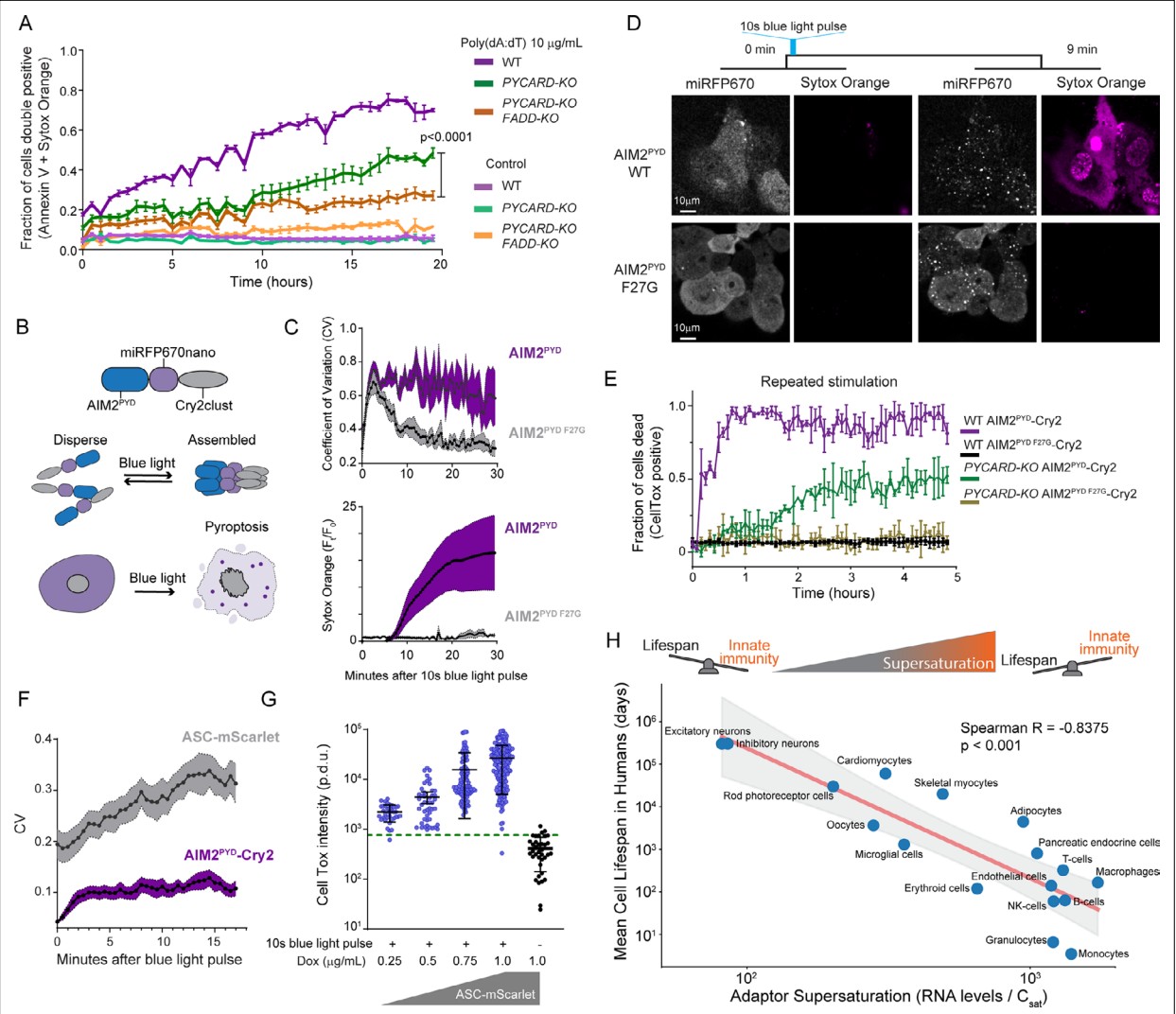

**Figure 4.** Innate immune adaptors are endogenously supersaturated. (**A**) Time course of apoptotic cell death of THP-1 cells following exposure to AIM2 ligand, poly(dA:dT). p-Value obtained from ANOVA followed by pair comparison. (**B**) Schematic diagram of the experiment to transiently optogenetically stimulate AIM2$^{PYD}$ to monitor ASC$^{PYD}$ assembly. This experiment was conducted in HEK293T cells because they do not undergo pyroptosis. (**C**) Top, Time course of fluorescence intensity distribution in THP-1 cells following 10 s of optogenetic activation, showing that WT AIM2$^{PYD}$ forms clusters (high CV) that persist and induce cell death, while the F27G solubilizing mutant (*Lu et al., 2014*) forms clusters that subsequently disperse. Bottom, normalized Sytox Orange fluorescence intensity for the experiment in the top panel. (**D**) Representative confocal microscopy images from a timelapse of THP-1 monocytes showing that transient optogenetic stimulation of WT but not F27G mutant of AIM2$^{PYD}$ causes it to form puncta that coincide with cell death. Sytox Orange was used for this experiment because it can be excited without activating Cry2. Scale bar 10 µm. (**E**) Time course of cell death of THP-1 cells when subjected to a blue light pulse every 5 min ('repeated'), showing rapid cell death (violet trace) only when AIM2$^{PYD}$ is WT and when ASC is present. The absence of ASC results in slower death (green trace), consistent with apoptosis. The F27G mutation of AIM2$^{PYD}$ blocks cell death irrespective of ASC (black and golden traces). (**F**) Coefficient of variation (CV) of fluorescence distribution of AIM2$^{PYD}$-Cry2 and ASC-mScarlet-I in THP-1 *PYCARD-KO* cells following a 10 s blue light pulse. This shows that AIM2$^{PYD}$ and ASC-mScarlet-I (with slightly delayed kinetics) rapidly form clusters that persist well after stimulus removal. ASC-mScarlet-I was induced to only ~20% of the ASC expression in WT cells using 1.0 µg/mL doxycycline (dox). (**G**) Quantification of CellTox staining in individual ASC-mScarlet-I THP-1 *PYCARD-KO* cells 30 min after a 10 s blue laser pulse, at different levels of dox-induced ASC-mScarlet-I expression. Green dotted line indicates 95% confidence interval (CI) for background fluorescence intensity, above which cells were considered CellTox-positive. Error bars denote standard deviation. Control, n=37. 0.25 µg/mL dox, n=36. 0.5 µg/mL dox, n=47. 0.75 µg/mL dox, n=113. 1 µg/mL dox, n=180. (**H**) Top: The metastability of supersaturation implies that cells will occasionally inflame and/ or die from stochastic (without D/PAMPs) DFD nucleation, which creates a tradeoff between innate immunity and life expectancy. Bottom: Scatter plot showing the relationship between geometric mean of adaptor supersaturation including ASC, FADD, BCL10, TRADD, MAVS (as approximated by the ratio of transcription levels and C$_{sat}$ values) and mean lifespan for each cell type in the human body for which data is available (*Sender and Milo, 2021*) Cell types with greater DFD supersaturation have shorter mean lifespans. The red line represents the best-fit power-law regression, obtained by

*Figure 4 continued on next page*

(*Figure 5C*). These interactions are consistent with the residual cell death we observed in the *PYCARD FADD* double knockout, as well as previously observed crosstalk between these pathways and the close phylogenetic relationship between PYDs and DEDs (*Roncaioli et al., 2023*; *Bedoui et al., 2020*; *Gullett et al., 2022*; *Park et al., 2007b*).

On the whole, the observed network of nucleating interactions reveals that DFDs from different pathways generally function independently of each other, allowing adaptors to serve as orthogonal energy reservoirs for their respective signalosomes (as schematized in *Figure 2E*).

### DFD nucleation barriers are deeply conserved

Animals acquired DFDs horizontally from bacteria (*Aravind et al., 2024*; *Dalrymple and Jenkins, 1951*; *Kibby et al., 2023*), giving rise to an ancestral DISC (conserved across metazoa) and later inflammasome (conserved across vertebrates; *Figure 6A*; *Yuen et al., 2014*; *Sakamaki et al., 2014*). To investigate conservation of nucleation barriers, we characterized by DAmFRET the phase behaviors of DFD-containing constituents of a basal inflammasome – NLRP3, ASC, and CASP1 from zebrafish (*Danio rerio*); and a basal DISC – A0A1X7U321, FADD, and CASP8 from the model sponge, *Amphimedon queenslandica*. As for their human counterparts, the fish and sponge adaptors – but not the receptors and effectors – were supersaturated (*Figure 6B* and *Figure 6—figure supplement 1A–B*). These results suggest that the function of DFDs as energy reservoirs preceded the radiation of animals.

## Discussion

Our systematic investigation revealed that metastable supersaturation of a select subset of adaptor proteins provides an energetic basis for DFD-mediated signal amplification.

Seminal studies on the structures of DFDs have led to the common view that DFDs generally function as homotypic interaction modules (*Huoh and Hur, 2022*; *Ferrao and Wu, 2012*). We were therefore surprised to find that approximately half (51) of human DFDs failed to detectably self-assemble even when expressed well-beyond physiological concentrations. This suggests that a large fraction of DFDs function through hetero- rather than homo-oligomerization. From known examples, we postulate these will involve (1) receptors templating adaptor nucleation, (2) effectors binding to adaptor polymers, and (3) regulators repressing signaling by competing with 1 and 2 (*Fu et al., 2016*; *Wu et al., 2024*; *Lu et al., 2016*; *de Almeida et al., 2015*). Importantly, and despite a general appreciation that DFD polymerization functions to amplify signaling, most of the DFDs that self-assembled (36 of 57) lacked sufficient nucleation barriers to supersaturate at the cellular level. This precludes them from amplifying signals without additional energy expenditure by the cell. The only DFDs to supersaturate in full-length protein contexts were adaptors of inflammatory signalosomes. The sparsity of this property belied its functional importance, however, as indicated by higher centrality measures. Generally, the more central proteins of networks tend to be essential (*Jeong et al., 2001*), more abundant (*Nithya et al., 2023*), slower evolving (*Pang et al., 2016*), and more frequently targeted by pathogens than noncentral proteins (*Nithya et al., 2023*; *Schleker and Trilling, 2013*). Examining the topology of DFD networks revealed a core set of signal amplifying adaptors each interacting with a sequence-specified set of receptors, effectors, and regulators that categorically lack nucleation barriers. Because nucleation involves only a tiny fraction of the supersaturated adaptors, they are poised to amplify minute signals from their cognate receptors and transduce them to their cognate effectors, precisely as required for innate immunity. Furthermore, because this attribute is conserved across homologous animal adaptors, energy storage for on-demand signaling may be an ancestral function of DFDs.

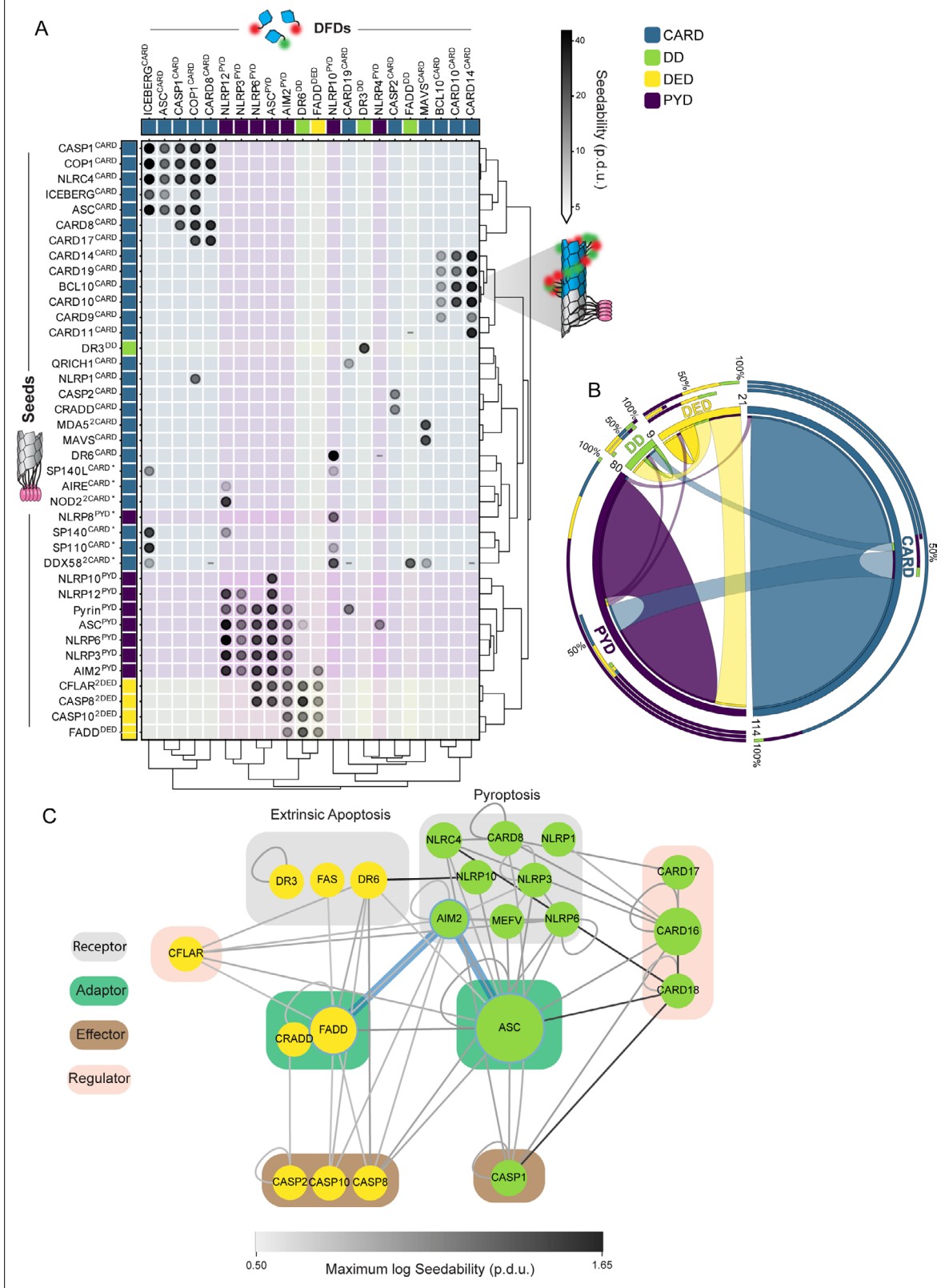

**Figure 5.** The nucleating interactome is highly specific. (**A**) Matrix of all nucleating interactions (gray-shaded circles) detected in a comprehensive DAmFRET screen of >10,000 DFD pairs. Each DFD-mEos (columns) was separately expressed with each DFD-µNS seed (rows). Darker shading of the circle denotes increased seedability. Interactions among members of the same signaling pathway (in legend) appear in color-shaded squares. Asterisk denotes seeds that were screened in a separate experiment from the rest. The matrix was clustered on seedability values, on a log scale, using the

*Figure 5 continued on next page*

*Figure 5 continued*

SciPy.cluster.hierarchy v1.11.1 linkage and dendrogram Python packages, using the Ward variance minimization algorithm to calculate distances. Procedure-defined units (p.d.u.). (**B**) Circos plot of the nucleating interactions summarized by DFD subfamily. Each subfamily is represented with a segment proportional to the number of DFDs with a nucleating interaction, as indicated by ribbons within and between segments. Inner stacked bars around the perimeter show the numbers of DFDs in each subfamily seeded by the subfamily in that segment. Middle stacked bars around the perimeter show the numbers of DFDs in each subfamily that seed the subfamily in that segment. Outer stacked bars around the perimeter show total nucleating interactions involving the subfamily in that segment. (**C**) Nucleating interactions involving DFDs in extrinsic apoptosis and pyroptosis, with blue edges highlighting the direct nucleating effect of AIM2 on FADD and ASC that is explored in *Figure 4*. The network was created in Cytoscape with node size corresponding to betweenness centrality and grouped by reported function. Interactions between FL proteins (*Supplementary file 2*) were included. Edge darkness indicates the seedability score of the corresponding interaction.

The online version of this article includes the following figure supplement(s) for figure 5:

**Figure supplement 1.** Generation and validation of the DFD nucleating interactome.

If supersaturation is indeed important for innate immune signaling, why are so few DFDs supersaturable? The answer may lie in the fact that each supersaturated node in a death-inducing pathway imposes a risk of unintentional death. We speculate that evolution therefore minimizes the number of supersaturated DFDs by restricting them to only the central nodes of the network. In this way, a restricted number of supersaturable DFDs can be continuously 'repurposed' with receptor and effector proteins specific to each D/PAMP and mode of cell death.

Two prominent inflammatory signaling adaptors – MyD88 and RIPK1 – defied the broader trend in their inability to supersaturate. Intriguingly, however, each functions in series with other adaptors that *do* supersaturate. MyD88 typically functions downstream of TIRAP, whose TIR domain drives nucleation-limited polymerization in vitro and in cells (*Ve et al., 2017*; *Lannoy et al., 2023*). Similarly, RIPK1 acts in conjunction with TRADD, leading to either NF-κB activation, apoptosis (via FADD), or necroptosis (via RIPK3). TRADD and FADD were shown here to supersaturate, while RIPK3 contains an amyloid-forming motif (RHIM) whose self-assembly is required for necroptosis (*Chen et al., 2022*). Hence, all of the partner adaptors of MyD88 and RIPK1 are either demonstrably or plausibly supersaturable, thereby obviating such functionality in MyD88 and RIPK1. Consequently, these exceptions in fact support the generalization that each inflammatory pathway has approximately one supersaturable node. Alternatively, MyD88 may be a false negative in our experiments, perhaps due to an absence of a required factor in yeast cells or the presence of a nucleating factor that does not exist in human cells. Unlike other adaptors which are effectively monomeric prior to activation, MyD88 forms discrete oligomers that activate by clustering into larger assemblies (*Moncrieffe et al., 2020*; *Cao et al., 2023*; *Fisch et al., 2024*). DAmFRET may be unable to resolve the transition between these multimeric forms. MyD88 was previously found to exhibit a nucleation barrier in vitro (*O'Carroll et al., 2018*), and to form switch-like and persistent assemblies at the subcellular level in vivo (*Fisch et al., 2024*; *Deliz-Aguirre et al., 2021*).

We liken supersaturated DFD adaptors to 'phase change materials' in industrial thermal batteries, accumulating potential energy for subsequent deployment upon nucleation (*Lizana et al., 2022*; *Sharma et al., 2009*). This differs fundamentally from signaling cascades that rely on chemical fuels like ATP. While ATP is a common good shared across cellular processes, DFD batteries are autonomous power sources, exclusive to each signaling pathway and therefore insulated from other cellular processes. This could allow for signal transduction independently of variable or compromised cell metabolism. Just as a battery cannot recharge itself, the DFD assemblies are effectively irreversible, committing the cell to terminal responses like death. This battery function rationalizes DFD prevalence over more common assembly modes like liquid-liquid phase separation (LLPS; *Figure 6C*). While LLPS offers sensitivity that can enhance D/PAMP detection (*Shen et al., 2021*; *Liu et al., 2023*), the sharp drop in its nucleation barrier with increasing supersaturation (*Vekilov, 2012*; *Martin et al., 2021*; *Shimobayashi et al., 2021*) makes it unsuitable for long-term, high-energy storage needed for explosive signal amplification through rapid assembly.

Signalosome effectors activate through simple induced-proximity mechanisms that lack specific structural requirements of the DFDs themselves (*Rodriguez Gama et al., 2022*; *Shkarina et al., 2022*; *Salvesen and Dixit, 1999*; *Lichtenstein et al., 2025*), which suggests that the structures of DFD polymers are determined by a function of the solution phase *prior to activation*. Our finding that the function is energy storage rationalizes the strikingly regular structure of DFD polymers. The evolution

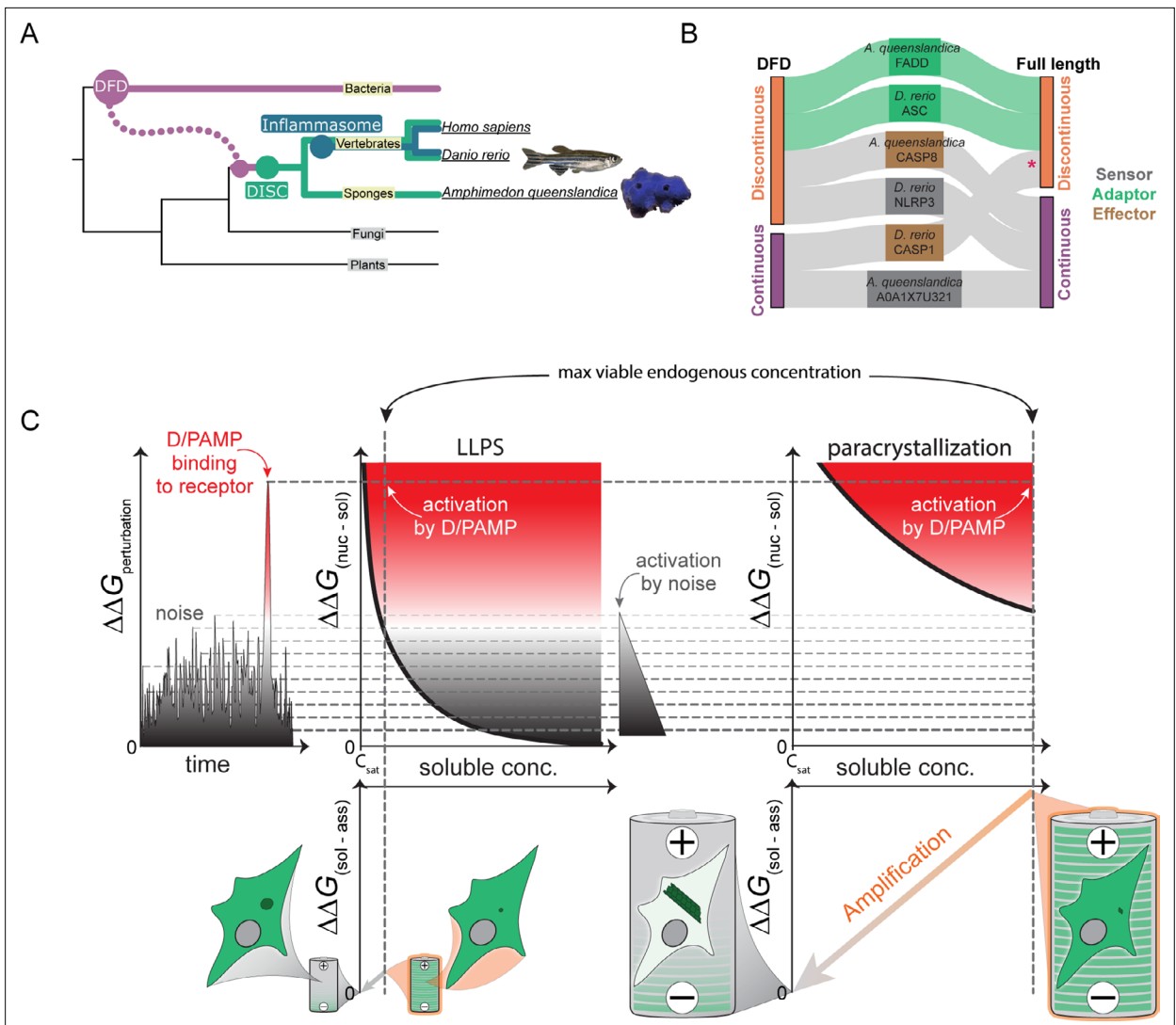

**Figure 6.** DFD nucleation barriers are deeply conserved. (**A**) Phylogenetic tree illustrating evolutionary relationships between DFD signaling pathways from bacteria to humans. (**B**) DAmFRET classifications for DFD-only and FL components of the DISC from the model sponge, *Amphimedon queenslandica*, and of the inflammasome from the model fish *Danio rerio*, showing that adaptors are specifically supersaturable. *$D. rerio$ CASP1$^{FL}$ exhibits a high $C_{sat}$ in the mid-micromolar range, *$D. rerio$ CASP1$^{FL}$ exhibits a high $C_{sat}$ in the mid-micromolar range based on prior calibrations of DAmFRET plots (**Khan et al., 2018**), which greatly exceeds the nanomolar concentration expected for endogenous procaspase-1 (**Walsh et al., 2011**), making it unlikely to supersaturate at endogenous concentrations. (**C**) Physical logic of DFD function. **Left:** Cells experience thermodynamic perturbations either from stochastic fluctuations (noise) or D/PAMP binding to innate immune receptors. These perturbations can nucleate supersaturated signaling proteins (dashed horizontal lines) with a probability that depends on the type of phase transition and specifically, whether it is accompanied by structural ordering. **Middle:** For phase separation in the absence of structural ordering (LLPS), the nucleation barrier ($\Delta\Delta G_{(nucleus - solute)}$) declines sharply with concentration beyond $C_{sat}$ (**Martin et al., 2021**; **Falahati and Haji-Akbari, 2019**), which increases its susceptibility to noise. This limits the level of supersaturation that can be maintained by a cell (vertical dashed line), and therefore, the extent to which assembly ($\Delta\Delta G_{(solute -- assembly)}$) can power signal amplification (tiny battery schematic). **Right:** For phase separation with structural ordering (paracrystallization as in adaptor DFD assemblies), the dependence of nucleation on concomitant intramolecular fluctuations buffers the barrier against concentration (as indicated by a shallower curve relative to LLPS), which allows cells to maintain much higher levels of supersaturation (**Khan et al., 2018**; **Buell, 2017**). Following nucleation, the assemblies grow and deplete soluble protein until it is no longer supersaturated, driving amplification (diagonal orange arrow) through proximity-dependent effector activation. The intrinsic nucleation barriers encoded by solution phase DFD ensembles therefore allow them to function as phase change batteries (giant battery schematic) to power innate immune signal amplification.

The online version of this article includes the following figure supplement(s) for figure 6:

**Figure supplement 1.** Demonstration of conserved energy storage capacity of DFDs.

of this function would have been driven by selection *against* premature assembly, favoring paracrystalline ordering whose infrequent nucleation allows for energy storage over a cell's lifetime. This differs from selection *for* a specific functional polymer structure, as seen for example in microtubules. This principle may also explain the functional substitution of DFDs with alternative paracrystalline modules like TIR domains and amyloids in some innate immune signaling pathways (*Kobe et al., 2025*), as well as the dearth of non-nucleated polymers, such as those of SAM domains, in innate immune signaling despite their prevalence in other signaling pathways (*Bienz, 2020*).

Despite functional nucleation barriers potentially driving the evolution of DFD polymerization, we emphasize that nucleation barriers are directly determined by the soluble ensembles that precede polymers. Our finding that only a subset of polymerizing DFDs exhibit nucleation barriers underscores this fact. These non-nucleation-limited polymers presumably have larger energetic differences between their orthogonal interfaces than those of nucleation-limited polymers, which would be expected to reduce the cooperativity of nascent polymer formation (*Heo and Chen, 2014*). Fully elucidating the DFD structure-function relationship will require future exploration of the conformational preferences of monomers and early stage oligomers.

The ability of a protein fold to crystallize has few structural constraints, allowing relatively small changes in sequence to produce orthogonal signaling modules. The resulting evolvability may be essential in the never-ending arms race against pathogens. High specificity also insulates pathways from each other and from cellular processes and metabolic fluctuations that could aberrantly activate them to lethal consequence (*Barton and Sontag, 2013*; *Capra et al., 2012*; *Zarrinpar et al., 2003*). The precise determinants of specificity will differ for each interaction, but prior work has uncovered principles that will likely prove general, such as the shape and electrostatic complementarity of interfaces and cross-compatibility of the polymers' helical architectures (*Lu et al., 2014*; *Wu et al., 2024*; *Garg et al., 2025*; *Park et al., 2007a*; *Li et al., 2018*).

Our findings imply that cells perpetually await death. The theoretical cumulative certainty of stochastic nucleation over time appears to be reflected in the observed relationship of DFD supersaturation to mortality rates across human cell types. We speculate that this underpins a fundamental tradeoff between innate immunity and life expectancy, potentially contributing to age-related inflammation and stem cell exhaustion (*López-Otín et al., 2023*).

### Limitations of the study

Multiple DFDs were found to populate stable ordered polymers in all cells. Although we did not observe a nucleation barrier for these, it is possible that they are supersaturable in vivo at much lower than the micromolar concentrations surveyed by DAmFRET. Similarly, a small number of DFDs that populated only a low AmFRET state did not express to high concentrations in yeast, and it is possible that they can self-assemble under cellular contexts that allow them to reach higher concentrations.

Our study necessarily simplifies and abstracts human innate immune signaling. We considered the cytoplasm of living yeast cells to be a suitable proxy for the physiological context of DFDs *as a whole*, while acknowledging that the endogenous contexts of DFDs differ from that in our experiments. For example, several of the DFDs normally localize to membranes or the nucleus in human cells, and this could impact their phase behavior. Nevertheless, we find no evidence that such localizations would reduce supersaturation, as none of the DFDs from nuclear proteins exhibited nucleation barriers in our experiments. Likewise, in the case of MAVS, which normally localizes to the mitochondrial membrane, we found that the FL protein (containing its mitochondrial localization signal) was more supersaturable than its DFD alone, suggesting that localizing the protein to the mitochondrial surface increases the nucleation barrier despite restricting its diffusional entropy and increasing its local concentration. Our work also does not explore the many ways that DFD assembly can be regulated transcriptionally and post-translationally. For example, NLRP3 is upregulated and becomes phosphorylated in response to LPS 'priming' of the inflammasome (*Song et al., 2017*), and polyubiquitination of RIG-I promotes its activation of MAVS (*Gack et al., 2007*). Such modifications imply that some signal amplification can occur upstream of supersaturated adaptors. The level of adaptor supersaturation differs between cell types (as we have shown) and is also likely to be dynamically regulated.

## Methods

### Key resources table

| Reagent type (species) or resource | Designation | Source or reference | Identifiers | Additional information |
|---|---|---|---|---|
| Strain, strain background (*Saccharomyces cerevisiae*, S288c) | rhy1713 | PMID:29979963 | | |
| Strain, strain background (*S. cerevisiae*, S288c) | rhy2153 | PMID:35727133 | | |
| Strain, strain background (*S. cerevisiae*, S288c) | rhy2977 | PMID:36920097 | | |
| Strain, strain background (*S. cerevisiae*, S288c) | rhy3078a | PMID:37921648 | | |
| Cell line (*Homo sapiens*) | HEK293T | American Type Culture Collection | RRID:CVCL_0063 | |
| Cell line (*H. sapiens*) | THP-1 | American Type Culture Collection | RRID:CVCL_0006 | |
| Cell line (*H. sapiens*) | THP-1 PYCARD-KO | Invivogen | RRID:CVCL_A8AN | |
| Antibody | anti-Actin (Mouse monoclonal) | Santa Cruz Biotechnology | RRID:AB_626630 | Protein Simple (1:50); Western Blot (1:1000) |
| Antibody | anti-PYCARD (Mouse monoclonal) | Santa Cruz Biotechnology | RRID:AB_2737351 | Protein Simple (1:200); Western Blot (1:1000) |
| Antibody | anti-FADD (Mouse monoclonal) | Sigma-Aldrich | RRID:AB_2100627 | Protein Simple (1:10); Western Blot (1:500) |
| Commercial assay or kit | Sytox Orange | ThermoFisher | S11368 | 1:1000 |
| Commercial assay or kit | Annexin V Alexa568 | ThermoFisher | A13202 | 1:200 |
| Commercial assay or kit | Annexin V Alexa488 | ThermoFisher | A13201 | 1:200 |
| Commercial assay or kit | Incucyte Caspase-3/7 Dye | Sartorius | 4440 | 1:1000 |
| Software, algorithm | Fiji/ImageJ | http://fiji.sc | RRID:SCR_002285 | |
| Software, algorithm | GraphPad Prism 9 | https://www.graphpad.com/ | RRID:SCR_002798 | |

## Reagents and antibodies

Hygromycin B (Invivogen, ant-hg-1), Penicillin-Streptomycin (Thermo Fisher, 1514014gp), PMA (BioVision, 1544–5), Puromycin (Invivogen, ant-pr-1), Sytox Orange (Thermo Fisher, S11368), CellTox (Promega, G8741), Annexin V Alexa568 (1:200, Thermo Fisher, A13202), Annexin V Alexa488 (1:200, Thermo Fisher, A13201), Incucyte Caspase-3/7 Dye (Sartorius, 4440). Antibodies, anti-ASC (Santa Cruz Biotechnology, sc-514414), anti-FADD (Sigma, 05–486), anti-Actin (Santa Cruz Biotechnology, sc-8432) were obtained from the indicated vendors.

## Structural analyses

Twelve human proteins contain two DFDs, typically one closely following the other. To determine if the DFDs in such pairs should be evaluated independently or together, we used the predicted aligned error (PAE) matrix generated by AlphaFold3 (*Varadi et al., 2022*). PAE is the expected positional error at residue x if the predicted and actual structures are aligned on residue y. Seven of the DFD pairs exhibited very low interdomain PAE scores comparable to those of the component DFDs (*Figure 1— figure supplement 1B* bottom), suggesting a conserved fixed geometric relationship between the domains. We therefore considered these tandem DFDs as single members of their respective subfamilies. Similarly, we excluded the annotated (*Wu et al., 2025*). PYD-like domain of CENP-N because the PAE matrix and experimental structures show that it is in fact part of a larger non-DFD.

## Plasmid construction

Yeast expression plasmids were made as previously described (*Khan et al., 2018*). Briefly, we used a high copy episomal vector, V08, which contains inverted BsaI sites to support Golden Gate cloning, followed by a rigid helical linker 4 x(EAAAR) and mEos3.1 ('mEos'). This vector drives the expression of proteins from a *GAL1* promoter and contains the auxotrophic marker *URA3*. The vector V12 is identical to V08 except that mEos and linker precede rather than follow the BsaI sites, for expressing proteins

with an N-terminal fusion. Inserts were ordered as yeast codon-optimized GeneArt Strings (Thermo Fisher) flanked by Type IIs restriction sites for ligation between BsaI sites in V08 and V12. Fusions were made opposite the native N- or C-terminus of each DFD to minimize non-native steric effects. All other inserts were cloned into respective vectors via Gibson assembly between the promoter and respective tag. All plasmids were verified by Sanger sequencing. All expression plasmids are listed in *Supplementary file 1*.

Lentivirus vectors were as previously described (*Rodriguez Gama et al., 2022*). Briefly, optogenetic constructs were cloned into pLV-EF1a-IRES-Hygro (Addgene #85134), which encodes a hygromycin B resistance cassette. To create lentiviral vectors expressing the optogenetic constructs fused with miRFP670nano (*Oliinyk et al., 2019*) and Cry2, the corresponding sequences of AIM2[PYD], APAF1[CARD], and NLRC4[CARD] were inserted via Gibson assembly into pLV-EF1a-IRES-Hygro. Finally, the doxycycline-controlled lentiviral vectors were cloned via Gibson assembly with the respective coding sequences from PYCARD, CASP9, CASP1, and mScarlet-I into pCW57.1 (Addgene #41393). All lentivirus vectors are listed in *Supplementary file 3*.

## Yeast strain construction

Unseeded DAmFRET experiments were conducted using *S. cerevisiae* strains rhy1713, rhy2977, and rhy3078a (*Khan et al., 2018*; *Kandola et al., 2023*; *Miller et al., 2023*). To create strains expressing DFD seeds, we first transformed AseI digests of each DFD plasmid along with a plasmid expressing Cas9 and a guide RNA targeting the *URA3* markers into rhy2153 (*Rodriguez Gama et al., 2022*). This strain contains a genomic landing pad consisting of natMX followed by the *tetO7* promoter and counterselectable *URA3* ORFs derived from *C. albicans* and *K. lactis*, and stop-µNS-mCardinal as described (*Rodriguez Gama et al., 2022*). Successful integration of the insert replaces the *URA3* marker with the gene of interest and fuses to the protein's C-terminus µNS-mCardinal, under the control of a doxycycline-repressible promoter. Transformants were selected for resistance to 5-FOA and validated for successful seed integration by detection of mCardinal expression using flow cytometry. The arrayed library of resulting strains was then mated to each of the rhy1713 strains expressing separate DFD-mEos fusions, by pinning each pair of strains together onto agar omnitrays containing SD-URA+NAT + dox media. The resulting colonies were then pinned into liquid SD-URA+NAT + dox for continued diploid selection and creation of glycerol stocks. The entire nucleating interaction screening consisted of 384 96-well plates.

## DAmFRET assay preparation and data collection

We performed DAmFRET as previously described (*Khan et al., 2018*). Briefly, single transformant yeast colonies were inoculated in 200 µL of SD-URA in a 96-well microplate well and incubated in a Heidolph Titramax platform shaker at 30 °C, 1350 RPM overnight. Cells were washed with sterile water, resuspended in galactose-containing media, and allowed to continue incubating for approximately 20 hr. Microplates were then illuminated for 25 min with 320–500 nm violet light to photoconvert a fraction of mEos molecules from a green (516 nm) form to a red form (581 nm). At this point, cells were either used to collect microscopy data or continue the DAmFRET protocol.

For the nucleating interaction screen, glycerol stock plates were pinned into liquid SD-URA without dox and incubated for 16 hr at 30 °C with 1350 RPM shaking overnight. We then resuspended cells in fresh SD-URA media and continued incubation for an additional 20 hr. After this, we resuspended cells in SGal-URA and continued incubation for 20 hr to induce protein expression. Finally, we resuspended cells in fresh SGal-URA for 4 hr prior to DAmFRET data collection. The library was then consolidated into 96 384-well plates.

DAmFRET data were collected on a ZE5 cell analyzer cytometer. Autofluorescence was detected with 405 nm excitation and 460/22 nm emission; side scatter (SSC) and forward scatter (FSC) were detected with 488 nm excitation and 488/10 nm emission. Donor and FRET fluorescence were detected with 488 nm excitation and 425/35 nm or 593/52 nm emission, respectively. Acceptor fluorescence was detected with 561 nm excitation and 589/15 nm emission. For each well, we collected a volume of 13 µL, resulting in approximately 500,000 events per sample. Data compensation was done in the built-in tool for compensation (Everest software V1.1) on single-color controls: non-photoconverted mEos and dsRed2 (as a proxy for the red form of mEos). For nucleating interactions, we included an additional channel for mCardinal intensity with 561 nm excitation and 670/30 nm emission.

## DAmFRET data analysis

Data were processed on FCS Express Plus 6.04.0015 software (De Novo). Events were gated for single unbudded cells by FSC vs. SSC, followed by gating of live cells with low autofluorescence and positive donor and acceptor fluorescence. With the exception of TNFRSF10A[DD] (TRAIL-R1) (rhx2933), which failed to express with either its C- or N-terminus tagged, all expression plasmids were processed. Plots represent the distribution of AmFRET (FRET intensity/acceptor intensity) vs. acceptor intensity (protein expression).

We then analyzed the data as previously described (*Kandola et al., 2023*). Briefly, FCS files were gated using an automated R script running in flowCore. Before gating, the forward scatter (FS00.A, FS00.W, FS00.H), side scatter (SS02.A), donor fluorescence (FL03.A), and autofluorescence (FL17.A) channels were transformed using a logicle transform in R. Single cells were gated using FS00.A vs SS02.A and FS00.H vs FS00.W. These were gated for expressing cells using FL03.A vs FL17.A. Cells falling within these gates were then exported as FCS3.0 files for further analysis.

DAmFRET histograms were divided into 64 logarithmically spaced bins across a predetermined range large enough to accommodate all data sets. The upper gate values were determined for each bin as the 99th percentile of the DAmFRET distribution in that bin. We used the expression of mEos alone to delineate the region of a DAmFRET plot that corresponds to no assembly. For all samples, cells falling above this region are considered to contain protein assemblies (FRET-positive). The fraction of cells in the assembled population was plotted as a ratio to the total cells in the bin for all 64 bins. The gross fraction of such cells expressing a given protein is reported as fgate.

## Determination of continuity

We initially attempted to classify each DAmFRET dataset as one-state or two-state, and discontinuous or continuous for the latter, using an algorithm previously developed for this purpose (*Posey et al., 2021*). However, the algorithm invariably misclassified discontinuous datasets as 'continuous' when the AmFRET level of the high FRET state changed with concentration, that is exhibited positive or negative slope as for FAS[DD] and CASP2[CARD], respectively. Due to this limitation, we adopted a different method.

To determine the continuity of an adequately expressed plasmid, we analyzed the distribution of AmFRET values about the transition region. To do so, we fit a spline to the median AmFRET values across binned concentrations that met a density cutoff defined by a minimum of 20 cells, a minimum cell density of 500, and no more than 25% of cells being defined as an outlier. The bin density is measured by the number of cells divided by the interquartile range (IQR) of AmFRET values within that bin. The number of bins was determined using Scott's rule for number of bins in a histogram. The median AmFRET values were calculated for each bin and then denoised using the Python SciPy.signal. sosfiltfilt, resulting in the spline fit. This was bootstrapped 100 times with the average of all bootstraps reported as the final spline fit. The resulting spline was used to determine the transition region. The transition point is defined as the concentration with the greatest change in AmFRET. Therefore, this value was calculated as the maximum of the first derivative of the fitted spline, using the numpy.diff package. The transition range is defined as the region between the maximum positive and negative rate of change, indicating where the transition between FRET states is starting and ending, respectively. The transition starting point is calculated as the maximum of the second derivative that lies before the transition point, and the transition endpoint is calculated as the minimum of the second derivative that lies after the transition point. This is done on all 100 bootstrapped splines, with the median of each measure reported. Hartigan & Hartigan's dip test for unimodality was used to determine the continuity of AmFRET values within the determined transition range. Plots are classified as 'discontinuous' or 'continuous' for p-values less than or greater than 0.05, respectively.

Plasmids that were determined to have a continuous transition were further classified as uniformly low ('low'), uniformly high ('high'), or transitioned from low to high with increasing concentration ('low to high'). This was done by extracting the minimum and ending AmFRET values from the spline fit. As a reference for the AmFRET value that indicates the start of a high FRET state, the AmFRET value of the transition start point of a control plasmid, rhx0927, was used. For each plasmid, if its minimum and ending AmFRET values fell below the reference, it was classified as low. If both fell above, it was classified as high. Those with a minimum value below and an ending value above were classified as 'low to high'. The majority classification of replicates for each plasmid is reported.

## Calculating centrality measures

To determine betweenness and degree of centrality, we extracted interactions involving DFD-containing proteins (listed in *Supplementary file 1*) from STRING version 12.0 (*Szklarczyk et al., 2023*), considering only physical interactions with scores of 900 or higher. Using NetworkX v3.1, we analyzed these interactions as an undirected graph to calculate betweenness and degree of centrality. For proteins with multiple DFDs, we classified a protein as discontinuous if any of its DFDs were identified as discontinuous.

## Determination of positive nucleating interactions

We first excluded files with fewer than 2500 events positive for mEos or mean acceptor intensities less than 3.5 p.d.u. From this, any DFD or seed left with less than 25% of their original instances after filtering was removed from the analysis completely. Next, we identified nucleating interactions as DFD pairs that decreased the C50 and increased the fraction assembled (fgate). We standardized all variables for each experimental batch of DFDs to a mean of 0 and variance of 1. We then determined the outlier degree for C50 and fgate based on the number of interquartile ranges below or above the median for these values. This was done directionally on a per-DFD basis. We defined the 'nucleating interactions' for a given mEos-fused DFD as those whose mean of these two values (reported as 'seedability') is greater than or equal to 3 standard deviations above the mean of all seedability values. We confirmed that the seedability values for most DFD pairs partitioned with that of two negative controls included for each DFD.

To evaluate the reproducibility of our assay, we replicated it for a set of 36 DFDs. This replicate analysis mirrored the original, except it utilized the DFD distributions and cutoff values from the first set. The Pearson correlation (R) between the two sets was 0.91 (p<0.0001). To minimize the impact of random variations in negatives and outliers, we excluded double negative instances, resulting in a slightly altered Pearson correlation (R) of 0.90 (p<0.0001). Of the 3423 DFD +seed combinations reassessed, 16 showed inconsistent hit-calling, indicating an assay consistency rate of 99.53% with a 95% confidence interval ranging from 99.30% to 99.76%.

## Approximating $C_{sat}$ and supersaturability

To generate an average DAmFRET curve, we computed the mean of each histogram bin in the DAmFRET dataset, focusing on bins containing a minimum of 100 cells. The average DAmFRET curve for each DFD in the presence of its self-seed was fit to a Weibull function as follows. We first calculated the average AmFRET value in each concentration bin. The resulting curves resemble the fraction-assembled curves except that the asymptote is the maximum AmFRET value rather than 1. Therefore, we used the following equation for fitting:

$$AmFRET\left(c\right) = Amp\left[1 - exp\left(-ln\left(2\right)\left(\frac{c}{C50}\right)^{a}\right)\right] \tag{1}$$

Here Amp is the AmFRET asymptotic value of the curve, c is the concentration, $C50_{seeded}$ is the concentration at which the curve, in the presence of its self-seed, has reached 50% of its asymptotic value, and $a$ describes the steepness of the stretched exponential. Initial values of the parameters were chosen based on Gaussian smoothed versions of the curves and constrained in the fit to at minimum a twofold change from those initial guesses. The $a$ parameter was constrained between 0.1 and 10 based on expected reasonable values of that parameter. At low and high concentration values, the average AmFRET values are clearly unstable and influenced by noise and minor compensation errors. Therefore, we chose the beginning and ending points of each curve by visual inspection, choosing starting points where the curve begins to increase and ending points where the curve levels off. Error values were determined from Monte Carlo simulations as in the fitting of fraction assembly.

To approximate the supersaturability of DFDs, we used the ratio of the average C50 in the presence of the respective DFD's seed ($C50_{seeded}$) to the average C50 in the presence of a null-seed ($C50_{stochastic}$). Ratios were only calculated between C50s obtained from the same batch run.

## Cell culture

HEK293T cells and THP-1 cells were purchased from ATCC. THP-1 *PYCARD*-KO (thp-koascz) cells were purchased from InvivoGen. HEK293T cells were grown in Dulbecco's Modified Eagle's Medium

(DMEM) with L-glutamine, 10% fetal bovine serum (FBS), and PenStrep 100 U/mL. THP-1 cells were grown in Roswell Park Memorial Institute (RPMI) medium 1640 with L-glutamine and 10% FBS. All cells were grown at 37 °C in a 5% $CO_2$ atmosphere incubator. Cell lines were regularly tested for mycoplasma using the Universal mycoplasma detection kit (ATCC, #30–1012 K).

## Generation of stable cell lines

Stable cell lines were created as described (*Rodriguez Gama et al., 2022*). Briefly, constructs were packaged into lentivirus in a 10 cm plate 60% confluent of HEK293T cells using the TransIT-LT1 (Mirus Bio, MIR2300) transfection reagent and 7 µg of the vector, 7 µg psPAX2, and 1 µg pVSV-G. Lentivirus was harvested and incubated with 293T with polybrene or infected at 1000 × g for 1 hr for THP-1 cells. For transduction of pCW57.1 derived vectors, HEK293T and THP-1 cells were selected with Puromycin (1 µg/mL) for 7 days. After this time, cells were sorted for positive expression of mScarlet-I and expanded in continuing selection with puromycin. For transduction of plasmids encoding fusions to miRFP670nano-Cryclust, THP-1 and HEK293T cells were selected with hygromycin B (350 µg/mL and 150 µg/mL, respectively) for 7 days. Cells were sorted for positive expression of miRFP670nano and expanded for further experiments with continued selection. To generate THP-1 PYCARD-KO+FADD KO cells, sgRNA targeting FADD exon 1 was cloned into the lentiCRISPR v2-Blast (Addgene #83480). This vector was packaged into lentivirus as described above. THP-1 PYCARD-KO cells were transduced using spinfection and supplemented with polybrene. 24 hr after spinfection, media was replaced. 48 hr after spinfection, cells were selected with blasticidin (1 µg/mL). After 10 days of blasticidin selection, single-cell clonal expansion was done by serial dilution of resistant cells to achieve complete knockouts. Selected wells were analyzed by immunoblot to confirm the absence of FADD protein and sequence-verified.

## High-content imaging analysis

High-content imaging was performed on the Opera Phenix high-content screening system (PerkinElmer) using a 63 x water immersion objective. Briefly, yeast (rhy2977) transformed with individual plasmids were cultured and induced as for DAmFRET assays. Then, 10 µL were transferred into a well containing 90 µL of SGal-URA of a 96-well optically clear flat-bottom plate (PerkinElmer 6055302). Data analysis of the high content imaging was performed in Fiji. Images of mEos were acquired using 488 nm excitation and a standard GFP filter set. Small z-stacks were acquired over 5 µm total range with 1 µm steps. The image containing the brightest mEos signal was used. The mEos signal was then background-subtracted with a rolling ball radius of 100 pixels, then found and converted to Fiji ROIs using the Fiji Default method of traditional image thresholding. The mean, standard deviation, and aspect ratio (AR) were measured for each object. The coefficient of variation (CV) in pixel intensity was calculated for every object following the formula: CV = Std Dev/Mean*100. The wells were divided into three categories based on their AR and CV. Objects that had a CV >55 and an AR >1.159 were designated 'fibrillar'. Objects that had a CV >55 and an AR <1.16 were designated 'punctate'. Finally, objects that had a CV <55 and an AR <1.16 were designated 'diffuse'. These cutoffs were determined manually from a visual inspection of the data. Plasmids rhx4763 - rhx4767 were acquired in a second data set that used different laser powers and integration times. Hence, for these plasmids specifically, 'fibrillar' had a CV >17 and an AR >1.4, 'punctate' had a CV >17 and an AR <1.41, and 'diffuse' had a CV <18 and an AR <1.41. These cutoffs were determined manually from a visual inspection of the data. The results were then manually verified for all wells. Seven plasmids were inconsistently classified by these cutoffs. Of these, manual inspection confirmed 'diffuse' morphology for rhx1033 and rhx1133 and prompted reclassification of rhx2935, rhx2637, rhx1121, rhx1070, and rhx1055 from 'fibrillar' to 'punctate'. Plasmid rhx2232 was incorrectly classified as punctate, but images display fibrillar morphology. Three others (rhx1113, rhx1097, rhx2937) had anomalously high AR due to low expression. Representative microscopy images are included for all plasmids in our repository.

## Fluorescence microscopy and optogenetic nucleation

The yeast and mammalian cells were imaged in an LSM 780 microscope with a 63 x Plan-Apochromat (NA = 1.40) objective. T-Sapphire was excited with a 405 nm laser. mEos and mScarlet-I were excited with a 488 nm and 561 nm laser, respectively. For time-lapse imaging, samples were maintained at 37 °C and 5% $CO_2$ with a stage top incubator. To stimulate Cry2clust, we used the 488 nm laser at a

power setting of 50% for a pulse of 10 s, which is the amount of time it took to scan the user-generated region of interest unless indicated otherwise. 561 and 633 nm lasers were used for imaging mScarlet-I and miRFP670nano, respectively. Pyroptosis events were tracked by incorporating the Sytox Orange reagent into the cell. To quantify the CV, images were subjected to an in-house Fiji adapted implementation of Cellpose for cellular segmentation (*Stringer et al., 2021*). The Cellpose-generated regions of interest (ROIs) were used to measure specified imaging channels.

For quantification of cell death events using IncuCyte (Sartorius), THP-1 cells were plated on a 24- or 96-well plate at a density of $4x10^8$/well or $1x10^8$/well, respectively, with PMA (10 ng/mL) for 16 hr. Media was replaced with fresh media supplemented with Annexin V-Alexa488 and Sytox Orange (1:1000). For AIM2-Cry2clust activation, an initial collection of unexposed measurements was taken for 30 min. Then, the plate was exposed to a 488 nm laser every 5 min. For treatments with poly(dA:dT), cells were treated and immediately subjected to imaging every 30 min for 19 hr. Positive cells for either fluorophore were identified using the integrated software in the IncuCyte instrument.

For optogenetic activation of APAF1[CARD] and NCLR4[CARD], HEK293T cells expressing lentivirus constructs were seeded on a 35 mm dish (ibidi) at a density of $4x10^4$/mL with 2 mL of media. The next day, dox was added at a concentration of 1 µg/mL to induce the expression of mScarlet-I tagged proteins. 24 hr after protein induction, media was replaced with fresh media supplemented with IncuCyte Caspase-3/7 Dye (1:1000) 2 hr prior to the experiment or Annexin V-Alexa488. Cells were imaged using a spinning-disk confocal microscope (Nikon, CSU-W1) with a ×60 Plan Apochromat objective (NA = 1.40) and a Flash 4 sCMOS camera (Hamamatsu). A region of interest (ROI) was selected to induce optogenetic activation for indicated times using a 488 nm laser at 50% laser power for the indicated time ranging from a fraction of a second to 30 s. ROIs were generated by the Cellpose segmentation algorithm around each cell contour. These ROIs were then used to measure the area, mean, standard deviation, and integrated density of each cell on the 488 nm and 560 nm fluorescence channels.

## Protein immunodetection

We performed capillary-based protein immunodetection (Wes, ProteinSimple) as described (*Rodriguez Gama et al., 2022*). Briefly, protein lysates were prepared as per recommended manufacturer instructions to a final concentration of 1 µg/mL. An assay plate was filled with samples, blocking reagent, primary antibodies (1:50 dilution for anti-Actin, 1:200 dilution for anti-PYCARD), HRP-conjugated secondary antibodies, and chemiluminescent substrate. The plate was subjected to electrophoretic protein separation and immunodetection in the fully automated capillary system. The resulting data was processed using the open-source software Compass (https://www.proteinsimple.com/compass/downloads/) to extract the intensities for the peaks corresponding to the expected molecular weight of proteins of interest. For western blot, cells were centrifuged at 1000 x *g* for 5 min and resuspended in lysis buffer 50 mM Tris (pH 7.4), 137 mM NaCl, 1 mM EDTA, 1% Triton X-100, 10 mM DTT (dithiothreitol), cOmplete Protease Inhibitor (1 tablet / 10 mL) (Roche, 11697498001). Protein lysates were resolved on a NuPAGE 4 to 12%, Bis-Tris gel and transferred onto a PVDF membrane (IPVH00010, Millipore) using the Pierce Power Blotter (ThermoFisher). The membrane was blocked with 5% skim milk and incubated overnight with the antibodies: anti-Actin (1:1000, sc-8432), anti-FADD (1:500, 05–486) and anti-PYCARD (1:1000, sc-514414). Primary antibody was removed by several washes with TBS +0.1% Tween-20 and then subjected to incubation with secondary antibody (anti-mouse-HRP, 7076 S, Cell Signaling Technology). The detection of protein bands was then carried out using enhanced chemiluminescence (ECL; SuperSignal West Pico Chemiluminescent Substrate, Thermo Fisher, 34577). The chemiluminescent signal was acquired by placing the membrane in a film cassette and exposing it to X-ray film (Kodak) at varying durations in a darkroom. After exposure, films were developed using an automatic film processor.

## Semidenaturing detergent-agarose gel electrophoresis (SDD-AGE)

SDD-AGE was performed as previously described (*Khan et al., 2018*). Briefly, cells were lysed using a 2010 Geno/Grinder with bead-beating. Samples were prepared with 2% sarkosyl and separated in a 1.5% agarose gel with 0.1% SDS. The distribution of mEos-fused proteins was analyzed directly in the gel with a GE Typhoon Imaging System. Images were processed to remove background using a 250-pixel rolling ball, cropped, and contrast-adjusted.

## Quantification and statistical analysis

Two-sided Student's t-tests were used for significance testing unless stated otherwise for two-sample comparisons. The graphs represent the means ±SEM of independent biological experiments unless stated otherwise. Statistical analysis was performed using GraphPad Prism 9, Python, and R packages.

## Acknowledgements

We thank Nick Grishin and Lisa Kinch for creating structure-guided alignments of DFDs early in this work, and Mark Miller for assistance with illustrations. This work was performed to fulfill, in part, requirements for ARG's thesis research in the Graduate School of the Stowers Institute for Medical Research. This work was supported by the National Institute of General Medical Sciences (Award Number R01GM130927, to RH) and the National Institute on Aging (Award Number F99AG068511, to ARG) of the National Institutes of Health, the American Cancer Society (RSG-19-217-01-CCG to RH), and the Stowers Institute for Medical Research. The funders had no role in study design, data collection and analysis, or manuscript preparation. The content is solely the responsibility of the authors and does not necessarily represent the official views of the funders.

## Additional information

### Funding

| Funder | Grant reference number | Author |
|---|---|---|
| National Institute of General Medical Sciences | R01GM130927 | Randal Halfmann |
| National Institute on Aging | F99AG068511 | Alejandro Rodriguez Gama |
| American Cancer Society | RSG-19-217-01-CCG | Randal Halfmann |
| Stowers Institute for Medical Research | | Randal Halfmann |

The funders had no role in study design, data collection and interpretation, or the decision to submit the work for publication.

### Author contributions

Alejandro Rodriguez Gama, Conceptualization, Formal analysis, Funding acquisition, Investigation, Methodology, Writing – original draft, Writing – review and editing; Tayla Miller, Formal analysis, Investigation, Methodology, Writing – review and editing; Shriram Venkatesan, Conceptualization, Investigation, Methodology, Writing – review and editing; Jeffrey J Lange, Formal analysis, Investigation, Methodology; Jianzheng Wu, Xiaoqing Song, William D Bradford, Investigation; Malcolm Cook, Data curation, Visualization; Jay R Unruh, Formal analysis, Methodology; Randal Halfmann, Conceptualization, Supervision, Funding acquisition, Methodology, Writing – original draft, Writing – review and editing

### Author ORCIDs

Alejandro Rodriguez Gama https://orcid.org/0000-0003-3257-5549
Tayla Miller https://orcid.org/0009-0002-4424-7104
Shriram Venkatesan https://orcid.org/0000-0003-0778-2474
Jeffrey J Lange https://orcid.org/0000-0003-4970-6269
Jianzheng Wu https://orcid.org/0009-0005-0927-8919
William D Bradford https://orcid.org/0000-0001-8302-8305
Malcolm Cook https://orcid.org/0000-0002-2825-9183
Jay R Unruh https://orcid.org/0000-0003-3077-4990
Randal Halfmann https://orcid.org/0000-0002-6592-1471

Reviewer #1 (Public review): https://doi.org/10.7554/eLife.107962.3.sa1
Reviewer #2 (Public review): https://doi.org/10.7554/eLife.107962.3.sa2

Author response https://doi.org/10.7554/eLife.107962.3.sa3

## Additional files

### Supplementary files

Supplementary file 1. Metadata, construct specifics, and data for the human DFDs analyzed in this study, related to *Figures 1 and 2*. ND (not determined). NA (not applicable/available). FL (full-length). [a]rhx1073b, rhx1102b, rhx1108, and rhx1135 did not have a majority automated classification and were therefore classified by manual inspection. [b]Inspection of the DAmFRET plots of the six nonseedable discontinuous DFDs reveals that most are only slightly discontinuous and include CARD9[CARD], which we previously showed to have a negligible nucleation barrier (*Holliday et al., 2019*), indicating that at least some of these six are misclassified due to heterogeneity between cells around $C_{sat}$.

Supplementary file 2. Metadata, construct specifics, and data for additional proteins analyzed in this study, related to *Figure 2*. FL (full-length). [a]Mutations were made to inactivate enzymatic activity for cell death effectors.

Supplementary file 3. Specifics of lentivirus vectors used in this study, related to *Figures 4 and 5*.

Supplementary file 4. Nucleating interactome of DFDs, related to *Figure 5*. Note that plasmids rhx2938, rhx1071, rhx2934, rhx1113, rhx1064, rhx2933, and rhx1368; and seeds from rhx2938, rhx2933, and rhx2934; did not pass quality control as outlined in methods and were therefore omitted from the analysis.

Supplementary file 5. Transcript and protein abundance statistics, related to *Figure 2—figure supplement 1*.

MDAR checklist

### Data availability

All DAmFRET and representative microscopy data can be accessed interactively on a dedicated website https://simrcompbio.shinyapps.io/HalfmannLab-Nucleating_Interactome/. Original data underlying this manuscript can be accessed from the Stowers Original Data Repository at http://www.stowers.org/research/publications/libpb-2387.

The following dataset was generated:

| Author(s) | Year | Dataset title | Dataset URL | Database and Identifier |
|---|---|---|---|---|
| Gama AR, Miller T, Venkatesan S, Lange JJ, Wu J, Song X, Bradford D, Cook M, Unruh JR, Halfmann R | 2025 | Protein phase change batteries drive innate immune signaling and cell fate | http://www.stowers.org/research/publications/libpb-2387 | Stowers Institute, LIBPB-2387 |

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

## Appendix 1

### Supplementary materials

Supplementary text

Experiments to investigate the nature of DFD assembly. DAmFRET data do not clarify if self-assembly involves native DFD interactions rather than amyloid-like misfolding (*Khan et al., 2018*). To address this question, we introduced point mutations to disrupt assembly via conserved known interfaces between folded DFD subunits (*Lu et al., 2014*; *Holliday et al., 2019*). Across the multiple DFDs examined, all such mutations indeed reduced or eliminated the high-AmFRET population (*Figure 1— figure supplement 2A*). To directly evaluate the nature of DFD assembly in our experiments, we subjected the seeded and unseeded cells to semi-denaturing detergent-agarose gel electrophoresis (SDD-AGE), a technique that distinguishes amyloids from other protein states based on their detergent-resistance and size dispersity (*Halfmann and Lindquist, 2008*). We found that, unlike our amyloid control (RIPK1RHIM), none of the DFD assemblies survived sarkosyl exposure (*Figure 1— figure supplement 2B*), consistent with their retaining the death fold rather than misfolding into amyloid. Ongoing work is elucidating the physical basis of these nucleation barriers.

