## [Editor Report · eLife Assessment]

This **important** study investigates the self-assembly activity of all 109 human death-fold domains. The data collected using advanced microscopy and distributed amphifluoric FRET-based flow cytometry methods are **compelling** to support the "phase change battery" model that explains how signal amplification can occur without ATP consumption. This paper provides new insight into the thermodynamic control of protein phase behaviors within cells and will be of interest to those studying a variety of biological pathways involved in inflammatory responses and various forms of cell death.

---

## [Referee Report · Reviewer #1 (Public review)]

This is a high-quality and extensive study that reveals differences in the self-assembly properties of the full set of 109 human death fold domains (DFDs). Distributed amphifluoric FRET (DAmFRET) is a powerful tool that is applied here for a comprehensive examination of the self-assembly behaviour of the DFDs, in non-seeded and seeded contexts, and allows comparison of the nature and extent of self-assembly. The work reveals the nature of the barriers to nucleation in the transition from low to high AmFRET. Alongside analysis of the saturation concentration and protein concentration in the absence of seed, the work demonstrates that the subset of proteins that exhibit discontinuous transitions to higher-order assemblies are expressed more abundantly than DFDs that exhibit continuous transitions. The experiments probing the ~20% of DFDs that exhibit discontinuous transition to polymeric form suggest that they populate a metastable, supersaturated form, in the absence of cognate signal. This is suggestive of a high intrinsic barrier to nucleation.

The differences in self-assembly behaviour are significant and highlight mechanistic differences across this large family of signalling adapter domains, with identification of a small number of key supersaturated adapters, which exhibit higher centrality within networks, and can amplify signals and transduce them to effectors as required. The description of some supersaturated DFD adaptors as long-term, high-energy storage forms or phase change adaptors is attractive and is a framework that addresses many of the requirements for on-demand signaling and amplification in innate immunity. The identification of only a small number of key adaptors and high specificity suggests a mechanism for insulation of pathways from each other and minimisation of aberrant lethal consequences.

An optogenetic approach is applied to initiate self-assembly of CASP1 and CASP9 DFDs, as a model for apoptosome initiation in these two DFDs with differing continuous or discontinuous assembly properties. This comparison reveals clear differences in the stability and reversibility of the assemblies, supporting the authors' hypothesis that supersaturation-mediated DFD assembly underlies signal amplification in at least some of the DFDs. The study also reveals interesting correlations between supersaturation of DFD adapters in short- and long-lived cells, suggestive of a relationship between mechanism of assembly and cellular context. Additionally, the interactions are almost all homomeric or limited to members of the same DFD subfamily or interaction network and examination of bacterial proteins from innate immunity operons suggest that their polymerisation could be driven by similar mechanisms. Future detailed studies that probe the roles and activities of DFDs identified with continuous or discontinuous barriers to nucleation, through mutational analysis, in chimeric proteins and with high resolution studies of the assemblies, can build on this methodology and database.

The Discussion effectively places this work in the context of innate immunity effectors and adapters, explains and provides a justification of the phase change material analogy, and contrasts this mechanism with phase separation. The breadth and depth of the experimental investigations allow a new view of the role of nucleation barriers and supersaturation in DFD assembly and innate immunity pathways.

---

## [Referee Report · Reviewer #2 (Public review)]

This work studies the self-association behavior of 109 human Death Fold Domains (DFD) in eukaryotic cells and its connection to their function in innate immune signalosomes.

Using an amphifluoric FRET (DAmFRET) method previously developed by the authors, self-association is monitored as a function of protein concentration by Förster Resonance Energy Transfer in the cell.

Several DFDs are found to be in a supersaturable state and are considered energy reservoirs necessary for signal amplification.

The revised manuscript addresses most of the original concerns, resulting in a significant improvement.

The following observations are made:

(1) A group of DFDs shows a bimodal FRET distribution of no FRET and high FRET values at low and high protein concentration, which indicates a nucleation barrier. This conclusion is corroborated by the modification from a discontinuous to a continuous FRET transition by expressing a structural template or seed. The authors find that DFDs displaying discontinuous FRET behavior are supersaturated, and those that retain their discontinuous behavior in the context of the full-length protein correspond to protein adaptors of innate immune signalosomes.

(2) The authors indicate that the adaptors of inflammatory signalosomes act as energy reservoirs for signal amplification. This is not demonstrated, but it is assumed that the energy stored in the supersaturated state is released upon polymerization.

(3) This work also includes evidence showing that nonsupersaturable and supersaturable constructs of caspase-9 form puncta that dissolve or persist, respectively, upon apoptosome stimulation. The supersaturable construct also induces massive cell death, in contrast to the nonsupersaturable form. Although not demonstrated, these results could be related to the level of signal amplification.

(4) The cell's lifespan depends on the supersaturation levels of certain DFDs.

(5) Polymerization nucleated by interaction between DFDs from different pathways (different signalosomes) is rare.

(6) The study demonstrates the presence of nucleation barriers, inferred from supersaturable conditions, in the adaptor orthologs of zebrafish (*Danio rerio*) and the model sponge Amphimedon queenslandica, which indicates that this characteristic is conserved.

---

## [Author Response]

The following is the authors’ response to the current reviews.

Both reviewers indicated broad approval of the revised work, for which we are grateful.

Reviewer #1 requested no further changes.

Reviewer #2’s Public review states:

The authors indicate that the adaptors of inflammatory signalosomes act as energy reservoirs for signal amplification. This is not demonstrated, but it is assumed that the energy stored in the supersaturated state is released upon polymerization.

The “assumed” link between supersaturation and energy release is in fact a thermodynamic necessity. Supersaturation is, by definition, a high free energy state. Our data shows that triggering nucleation via optogenetics results in an immediate avalanche of polymerization and cell death. This is not an assumption; it is a direct observation of work performed by the system when the kinetic barrier is removed.

Reviewer #2 recommended:

Ideally, signal amplification could be tested by determining the levels of the final product, e.g., cytokines, activated caspases...

We did measure CASP3/7 activation, demonstrating a correlation with supersaturation of upstream adaptors. We do agree however that measuring the levels of other signaling products, including for each of the supersaturated pathways, would strengthen our claims. This will be the subject of future work.

The authors indicate a significant anticorrelation between the saturating concentrations and the transcript abundances (Figure 2B), reporting an R = -0.285.

This is correct… no change appears to be requested or warranted.

The following is the authors’ response to the original reviews.

**Public Reviews:**

**Reviewer #1 (Public review):**
Summary:This is a high-quality and extensive study that reveals differences in the self-assembly properties of the full set of 109 human death fold domains (DFDs). Distributed amphifluoric FRET (DAmFRET) is a powerful tool that reveals the self-assembly behaviour of the DFDs, in non-seeded and seeded contexts, and allows comparison of the nature and extent of self-assembly. The nature of the barriers to nucleation is revealed in the transition from low to high AmFRET. Alongside analysis of the saturation concentration and protein concentration in the absence of seed, the subset of proteins that exhibited discontinuous transitions to higher-order assemblies was observed to have higher concentrations than DFDs that exhibited continuous transitions. The experiments probing the ~20% of DFDs that exhibit discontinuous transition to polymeric form suggest that they populate a metastable, supersaturated form in the absence of cognate signal. This is suggestive of a high intrinsic barrier to nucleation.Strengths:The differences in self-assembly behaviour are significant and likely identify mechanistic differences across this large family of signalling adapter domains. The work is of high quality, and the evidence for a range of behaviours is strong. This is an important and useful starting point since the different assembly mechanisms point towards specific cellular roles. However, understanding the molecular basis for these differences will require further analysis.An impressive optogenetic approach was engineered and applied to initiate self-assembly of CASP1 and CASP9 DFDs, as a model for apoptosome initiation in these two DFDs with differing continuous or discontinuous assembly properties. This comparison revealed clear differences in the stability and reversibility of the assemblies, supporting the hypothesis that supersaturation-mediated DFD assembly underlies signal amplification in at least some of the DFDs.The study reveals interesting correlations between supersaturation of DFD adapters in short- and long-lived cells, suggestive of a relationship between the mechanism of assembly and cellular context. Additionally, the comprehensive nature of the study provides strong evidence that the interactions are almost all homomeric or limited to members of the same DFD subfamily or interaction network. Similar approaches with bacterial proteins from innate immunity operons suggest that their polymerisation may be driven by similar mechanisms.Weaknesses:Only a limited investigation of assembly morphology was conducted by microscopy. There was a tendency for discontinuous structures to form fibrillar structures and continuous to populate diffuse or punctate structures, but there was overlap across all categories, which is not fully explored.

We agree that an in-depth exploration of aggregate morphology would be interesting, but we feel it has limited relevance to the central findings of the manuscript. Our analysis established a relationship between discontinuous transitions and ordering based on the assumption that ordered assembly by DFDs involves polymerization, for which there is much precedent in the literature. Nevertheless, polymers of similar structure can form with different kinetics and hence, polymerization does not by itself imply an ability to supersaturate. We see this empirically in the “fibrillar” column in Fig. 1B. We have now elaborated this important point more fully in the relevant results section and in the discussion. Only five of the 108 DFDs in Fig. 1B warrant additional explanation. CASP4^CARD^ and IFIH1^tCARD^ lacked AmFRET but formed puncta; this could result from interactions with endogenous structures or condensates. DAPK1^DD^ and UNC5A^DD^ were classified as continuous (low) and fibrillar, but their AmFRET values are in fact higher than monomer control revealing that the fibrils simply comprise a small fraction of the protein. The puncta of UNC5A^DD^ additionally do not resemble the fibrillar puncta of other DFDs; we suspect it may be a false-positive resulting from localization to mitochondrial or other intracellular membranes. Finally, CASP2^CARD^ was inadvertently classified as punctate; this turns out to have been a technical artifact that has now been corrected (the fibrils wrapped around the cell perimeter to form ring-like puncta with anomalously low aspect ratios). We have now updated the methods section describing manual validation of our automated classification procedure, including which samples required reclassification. We have also now included all microscopy data in the public repository accompanying this manuscript.

The methodology used to probe oligomeric assembly and stability (SDD-AGE) does not justify the conclusions drawn regarding stability and native structure within the assemblies.

The reviewer is correct that SDD-AGE does not provide evidence against non-amyloid misfolding. It merely provides evidence that the DFDs are not forming amyloid (which are characteristically sarkosyl resistant). We have revised the sentence and further clarified that the distinction with amyloid specifically is important because amyloid is the only known form of ordered assembly (other than DFD polymers) with a nucleation barrier large enough to support deep supersaturation. Together with the series of interfacial mutants tested (and shown to impede assembly in all cases), the lack of sarkosyl-resistance provides evidence that the discontinuous DFDs are assembling through canonical DFD subunit interfaces.

The work identifies important differences between DFDs and clearly different patterns of association. However, most of the detailed analysis is of the DFDs that exhibit a discontinuous transition, and important questions remain about the majority of other DFDs and why some assemblies should be reversible and others not, and about the nature of signalling arising from a continuous transition to polymeric form.

We focused on discontinuous DFDs because this property allows for executive control over their respective pathways. They make signaling switch-like, which we argue is essential for innate immune responses. By contrast, and as illustrated in Figure 6D, supersaturation is required for a DFD to drive its own polymerization -- hence activation for a continuous DFD must be stoichiometrically coupled either with D/PAMP binding or positive feedback from downstream or orthogonal processes. We consider the principles underlying such regulation of signaling to be better established and understood than supersaturation, and hence built our narrative for this manuscript around the latter. Our original text addresses the fact that only a small fraction of DFDs are discontinuous. Specifically, this is expected in light of the fact that (a) only one supersaturated DFD is needed to make a signaling pathway switch-like, and (b) every supersaturated DFD renders the cell susceptible to spontaneous death. Evolution should therefore limit supersaturation to only the highly connected DFDs (i.e. adaptors), which is what is seen. In this view, the many nonsupersaturable DFDs have evolved to accessorize the central supersaturable DFDs with various sensor and effector modules. Our revised text attempts to further clarify this perspective.

Some key examples of well-studied DFDs, such as MyD88 and RIPK,1 deserve more discussion, since they display somewhat surprising results. More detailed exploration of these candidates, where much is known about their structures and the nature of the assemblies from other work, could substantiate the conclusions here and transform some of the conclusions from speculative to convincing.

We were likewise initially surprised about the inability of MyD88 and RIPK1 to supersaturate. We have now elaborated in the Discussion how our findings can be rationalized by the apparent supersaturability of other adaptors in MyD88 and RIPK1 signaling pathways. We additionally discuss prior evidence that MyD88 may indeed be supersaturable, and how our experimental system could have led to a false positive in the unique case of MyD88.

The study concludes with general statements about the relationship between stochastic nucleation and mortality, which provide food for thought and discussion but which, as they concede, are highly speculative. The analogies that are drawn with batteries and privatisation will likely not be clearly understood by all readers. The authors do not discuss limitations of the study or elaborate on further experiments that could interrogate the model.

We have now added to the discussion a section on the limitations of our study. We appreciate that our use of “privatisation” was confusing and have omitted it. However, we consider the battery analogy to accurately convey the newfound function of DFDs and anticipate that this analogy will ultimately prove valuable for biologists. To facilitate comprehension, we have now broadened our description of phase change batteries in the introduction.

**Reviewer #2 (Public review):**
Summary:The manuscript from Rodriguez Gama et al. proposes several interesting conclusions based on different oligomerization properties of Death-Fold Domains (DFDs) in cells, their natural abundance, and supersaturation properties. These ideas are:(1) DFDs broadly store the cell's energy by remaining in a supersaturated state;(2) Cells are constantly in a vulnerable state that could lead to cell death;(3) The cell's lifespan depends on the supersaturation levels of certain DFDs.Overall, the evidence supporting these claims is not completely solid. Some concerns were noted.Strengths:Systematic analysis of DFD self-assembly and its relationship with protein abundance, supersaturation, cell longevity, and evolution.Weaknesses:(1) On page 2, it is stated, "Nucleation barriers increase with the entropic cost of assembly. Assemblies with large barriers, therefore, tend to be more ordered than those without. Ordered assembly often manifests as long filaments in cells," as a way to explain the observed results that DFDs assemblies that transitioned discontinuously form fibrils, whereas those that transitioned continuously (low-to-high) formed spherical or amorphous puncta. It is unlikely to be able to differentiate between amorphous and structured puncta by conventional confocal microscopy. Some DFDs self-assemble into structured puncta formed by intertwined fibrils. Such fibril nets are more structured and thus should be associated with a higher entropic cost. Therefore, the results in Figure 1B do not seem to agree with the reasoning described.

The formation of microscopically visible elongated structures necessitates ordering on the length scale of 100s of nanometers. Otherwise surface tension would favor rounded aggregates. Conventional confocal microscopy is in fact well-suited and widely used to distinguish ordered from disordered assemblies in cells based on this principle.1,2 We are unaware of any examples of isolated DFDs forming regular polymers that manifest as round puncta or nets. The reviewer may be referring to full-length ASC, which forms a roughly spherical mesh of filaments because it has two DFDs joined by a flexible linker. This is not applicable to our analysis with single DFDs. Single DFDs polymerize in effectively one dimension; hence a spherical punctum formed by a single DFD can only happen through noncanonical interactions or clustering of small filaments, both of which reduce order relative to long filaments.

(2) Errors for the data shown in Figure 1B would have been very useful to determine whether the population differences between diffuse, punctate, and fibrillar for the continuous (low-to-high) transition are meaningful.

We have now performed two statistical analyses to address this. First, using Fisher’s exact test, we observe a highly significant association between the DAmFRET and morphology classifications (*p*-value: 0.0001). Second, to specifically address whether the continuous (low to high) category has a preferred morphology, we applied an Exact Multinomial Test using the total frequencies of each morphology. This test revealed that all categories are significantly enriched for particular morphologies, as now indicated in the figure and legend.

(3) A main concern in the data shown in Figure 1B and F is that the number of counts for discontinuous compared to continuous is small. Thus, the significance of the results is difficult to evaluate in the context of the broad function of DFDs as batteries, as stated at the beginning of the manuscript.

Fig. 1B simply reports the numerical intersections between fluorescence distribution classifications and DAmFRET classifications. In Fig. 1F, our use of the chi-square test is justified by a sufficiently large sample size. Nevertheless, we obtain similar results with Fisher's exact test that accounts for smaller sample size (Odds Ratio: 75.0, P-value: < 0.0001). See also our response to the related critique by Reviewer 1 regarding the small number of discontinuous DFDs.

(4) The proteins or domains that are self-seeded (Figure 1F) should be listed such that the reader has a better understanding of whether domains or full-length proteins are considered, whether other domains have an effect on self-seeding (which is not discussed), and whether there is repetition.

We define and consistently use “DFDs” to refer to domains, and “FL” or “DFD-containing protein” to refer to FL proteins. The Figure 1 title and corresponding section title both indicate the data refer to “DFDs”. The text callout for Figure 1F also directs readers to Table S1 where we believe the self-seeding results and details of constructs are clearly presented. There is no repetition. We have modified the legend to clarify that “Each DFD was co-expressed with an orthogonally fluorescent μNS-fused version of the same DFD.” We did not systematically evaluate seeding of FL proteins. We did however previously test self-seeding on seven representative FL proteins, and have now included those data in a new supplemental figure (S5). In short, only FL proteins with discontinuous distributions are self-seedable. These are limited to adaptors that had discontinuous seedable DFDs, revealing no adverse effect of FL protein context on seedability of adaptors (unlike receptors and effectors).

(5) The authors indicate an anticorrelation between transcript abundance and Csat based on the data shown in Figure 2B; however, the data are scattered. It is not clear why an anticorrelation is inferred.

An anticorrelation is indicated by the clearly placed negative R value at the top of the graph and the figure legend describing the statistical analysis.

(6) It would be useful to indicate the expected range of degree centrality. The differences observed are very small. This is specifically the case for the BC values. The lack of context and the small differences cast doubts on their significance. It would be beneficial to describe these data in the context of the centrality values of other proteins.

The *possible* range of centrality scores is 0 - 1, where 1 represents a protein interacting with every other protein in the network (degree centrality) or is on the shortest path between every other pair of proteins in the network (betweenness centrality). The *expected* range is difficult to address, as centrality values strongly depend on the size and function of the network. We considered that the SAM domain network could provide the most relevant comparison to the DFD network, as SAM domains resemble DFDs in size and structure, function heavily in signaling, are comparably numerous (76 in humans), and many of them form homopolymers (but importantly of a geometry that does not support nucleation barriers). We found that SAM domains have much lower betweenness centrality in their physical interaction network as compared to discontinuous DFDs (p = 0. 0003) while their degree centrality is not significantly different (Figure S3F). Nevertheless, we stress that what matters for our conclusion is that the continuous and discontinuous values are significantly different among DFDs. Since there is a large overlap in the distributions of centrality scores between the two classes of DFDs, we performed a more robust permutation test with the Mann Whitney U statistic and n = 10000. These tests reiterated that continuous and discontinuous DFDs have significantly different centrality scores (Degree centrality p = 0.008; Betweenness centrality p = 0.028) (Figure S3E).

(7) Page 3 section title: "Nucleation barriers are a characteristic feature of inflammatory signalosome adaptors." This title seems to contradict the results shown in Figure 2D, where full-length CARD9 and CARD11 are classified as sensors, but it has been reported that they are adaptor proteins with key roles in the inflammatory response. Please see the following references as examples: The adaptor protein CARD9 is essential for the activation of myeloid cells through ITAM-associated and Toll-like receptors. Nat Immunol 8, 619-629 (2007), and Mechanisms of Regulated and Dysregulated CARD11 Signaling in Adaptive Immunity and Disease. Front Immunol. 2018 Sep 19;9:2105. However, both CARD9 and CARD11 show discontinuous to continuous behavior for the individual DFDs versus full-length proteins, respectively, in contrast to the results obtained for ASC, FADD, etc.

We rigorously counter the inconsistent usage of the term “adaptor” in the signalosome literature by quantifying the centrality of each protein in the physical interaction network of DFD proteins. Such analysis shows that BCL10, which is also described as an adaptor, is the more central member of the CARD9 and CARD11 (CBM signalosome) pathways, and is therefore more “adaptor-like”. We have now elaborated this view in the text.

FADD plays a key role in apoptosis but shows the same behavior as BCL10 and ASC. However, the manuscript indicates that this behavior is characteristic of inflammatory signalosomes. What is the explanation for adaptor proteins behaving in different ways? This casts doubts about the possibility of deriving general conclusions on the significance of these observations, or the subtitles in the results section seem to be oversimplifications.

We agree that our initial presentation of these results and brief description of each protein’s function was insufficient to fully justify our conclusions. We have now elaborated that while FADD was historically considered an adaptor of extrinsic apoptosis, it is now appreciated as a pleiotropic molecule with both anti- and pro-inflammatory signaling functions. FADD’s pro-inflammatory roles include inflammasome activation and activating NF-kB through the FADDosome. We have now revised our section headings to avoid oversimplification.

(8) IFI16-PYD displays discontinuous behavior according to Figure S1H; however, it is not included in Figure 2D, but AIM 2 is.

We only tested a subset of FL proteins spanning different functions within diverse signalosomes. IFI16 was not included. Hence it could not be meaningfully included in Fig. 2D.

(9) To demonstrate that "Nucleation barriers facilitate signal amplification in human cells," constructs using APAF1 CARD, NLRC4 CARD, caspase-9 CARD, and a chimera of the latter are used to create what the authors refer to as apoptsomes. Even though puncta are observed, referring to these assemblies as apoptosomes seems somewhat misleading. In addition, it is not clear why the activity of caspase-9 was not measured directly, instead of that of capsae-3 and 7, which could be activated by other means.

We agree that describing our chimeric assemblies as “apoptosomes” could be misleading, and have now refrained from doing so. We measured caspase-3/7 instead of caspase-9 for purely technical reasons -- we were unable to find any reliable caspase-9 activity assays that were also compatible with our optogenetic and imaging wavelengths. In any case, our data with the widely used caspase3/7 reporter dyes confirm comparably effective signal propagation from the CASP9 versions to their relevant endogenous substrate for apoptotic signaling (pro-caspase-3/7). The subsequent differences in cell death efficiency between the two versions of CASP9 (Fig. 3E) cannot be attributed to indirect effects of blue light stimulation, because both versions received the same treatment. Note our stated justification for using these DFDs in the HEK293T background is that these cells lack NLCR4 and CASP1 proteins and therefore the activity we measure is due to the direct optogenetic activation.

The polymerization of caspase-1 CARD with NLRC4 CARD, leading to irreversible puncta, could just mean that the polymers are more stable. In fact, not all DFDs form equally stable or identical complexes, which does not necessarily imply that a nucleation barrier facilitates signal amplification. Could this conclusion be an overstatement?

Figure 3C shows that the polymers don’t simply persist following the transient stimulus -- they continue to grow. That is, the soluble protein continues to join the polymers for a net increase even though there is no longer a stimulus directing them to do so. This means the drive to polymerize is independent of the stimulus, i.e. the protein is supersaturated. In the absence of supersaturation, a difference in stability would simply change the rates at which the polymers shrink. That we see continued growth instead of shrinkage therefore cannot be explained just by a difference in stability. Nevertheless, the reviewer’s critique caused us to realize that increased persistence of the CASP1CARD polymers could contribute to signal amplification independently of supersaturation if they act catalytically (i.e. where each polymerized CASP9 subunit sequentially activates multiple CASP3/7 molecules), and we had not adequately considered this. Unfortunately, the relevant experimentalist has now moved on from the lab leaving us unable to conduct the necessary experiments to resolve these two effects in a timely fashion. Consequently, we have now tempered our interpretation of these data.

(10) To demonstrate that "Innate immune adaptors are endogenously supersaturated," it is stated on page 5 that ASC clusters continue to grow for the full duration of the time course and that AIM2-PYD stops growing after 5 min. The data shown in Figure 4F indicate that AIM2-PYD grows after 5 mins, although slowly, and ASC starts to slow down at ~ 13 min. Because ASC has two DFDs, assemblies can grow faster and become bigger. How is this related to supersaturation?

That AIM2-PYD assemblies appear to grow somewhat (although not significantly statistically) would be consistent with AIM2-PYD’s sequestration into the growing ASC clusters. All that matters for our conclusion regarding ASC is that ASC assemblies grow following cessation of the stimulus, which we now describe quantitatively. Supersaturation is defined as the ratio of total concentration to saturating concentration, which is an equilibrium property. For a given protein concentration, the presence of two DFDs, each contributing their own interactions to overall stability of the assembly, will increase supersaturation relative to the individual DFDs. Importantly, growth will not occur if the protein concentration lies below its C_sat_, no matter how many DFDs it has.

**Recommendations for the authors:**

**Reviewer #1 (Recommendations for the authors):**
It isn't clear what is implied by the final sentence of the Abstract. Some of the conclusions have a speculative tone and would be better described in less certain terms. The final sentence of the abstract should be omitted.

We have revised the abstract to add appropriate nuance but consider the final sentence to be both justified by our data and important to convey our findings to a broad audience.

How does the size and nature of the seed influence the outcome of these DFD interactions? Although some non-seeded experiments are described, the majority of the results are derived from seeded experiments. Further details about the seeds should be included. How is the size of the nucleus controlled, and will seeds of smaller or larger size generate the same pattern of results?

This is a very important question! The seeds comprised genetic fusions of each DFD to a condensate-forming domain, as described. While this system is insufficient to explore the size-dependence of nucleation, we are developing tools to do exactly that, for example our recently published multivalent nanobody against mEos3,[3] wherein we piloted its use to compare the size-dependence of ASC versus amyloid nucleation. Much further work will be needed to fully utilize this approach for the question of interest, and that is the subject of ongoing but open-ended work in the lab.

What is the implication of the observation that only ~20% of the DFDs exhibited a discontinuous transition from no to high AmFRET signal? Further discussion of the DFDs that exhibit a continuous transition would enrich the manuscript.

We consider the relationship to mortality important for understanding this observation. In the discussion we now explain that each supersaturated protein in a death-inducing pathway imposes a risk of unintentional death. We speculate that evolution therefore minimizes the number of supersaturated DFDs by restricting them to central nodes in the network. That way, a small number of supersaturable DFDs can be continuously “repurposed” with new receptor proteins for each D/PAMP. Additionally, as stated in our response to the related critique, we felt it was important to focus this manuscript on the novel concept of functional supersaturation necessarily at the expense of signaling regulation through better understood mechanisms.

Were the initial experiments with DFDs unseeded (Figure S1, F-G)? Clarify this in the text. The morphologies of all the subcellular assemblies appear similar. It is not possible to distinguish between long filaments and spherical or amorphous puncta (Figure S1F-G). Higher magnification images that allow evaluation and comparison of morphology should be provided.

The initial experiments were unseeded, as now clarified in the legend. We believe there was a misinterpretation resulting from both panels (S1F and G) showing fibrillar examples. To clarify, we have now added panel S1H showing representative DFDs classified as “punctate”, which we hope the reviewer agrees are clearly distinct from fibrillar.

The ASC and CARD14 assemblies in Figure S1G show very distinct fibrillar structures emerging from the mNS-DFD seeds. Please provide further explanation of the nature of these. Do these resemble ASC and CARD assemblies generated as a result of native stimuli rather than mNS-DFD seeds?

The μNS-DFD puncta contain numerous seeding competent sites, which presumably causes multiple fibrils to initiate and emanate from them. This and potential bundling of these fibrils produces the star-like shape. We have no reason to believe the internal structure of these fibers differs from native signalosome assemblies. For example, point mutations at native subunit interfaces that were previously shown to disrupt fibrilization and signaling likewise disrupt assembly in our DAmFRET experiments (Figure S2A). To our knowledge there exist no examples of high-resolution DFD fibril structures that were induced by native stimuli. However, recent work using super-resolution imaging confirmed that nigericin-triggered endogenous ASC specks comprise a network of filaments that superficially resembles our star-like assemblies.[4]

Figure S2B is presented as evidence that assembly is mediated by native-like interfaces rather than amyloid-like misfolding. These SDD-Age gels cannot be used to infer a native-like structure for the protein within the assemblies, only that the assemblies are (mostly) solubilised by incubation with sarkosyl. Many misfolding but non-amyloid-structure assemblies could be consistent with these results. Additionally, several of the samples appear to show insoluble aggregates within the wells, which could also be consistent with amyloid-type structures. What is the nature of these aggregates? Why is the NLRP3PYD sample so much more intense than the others? Why was FL-ZBP1 included when it does not contain a DFD? Why were no sarkosyl-resistant assemblies observed with RIPK3-RHIM when this is known to be highly amyloidogenic?

ZBP1 and RIPK3^RHIM^ were one of multiple proteins inadvertently included on the complete gel shown in the original figure that is not relevant to the manuscript; we have now spliced out these unnecessary lanes (indicated with dashed lines) to avoid confusion. We have found that the specific fragment of RIPK3^RHIM^ used in this experiment -- residues 446-464 -- does not allow for robust amyloid formation. We believe this is a steric artifact due to its small size (19 residues) relative to the fused mEos3, because a longer fragment (446-518) forms amyloid robustly. However the latter construct was not available at the time this experiment was done. Nevertheless, another known amyloid protein, RIPK1^RHIM^, does show the expected smears on this gel and suffices for the positive control for amyloid. We do not understand why the NLRP3^PYD^ sample is more intense than the others. However, this anomaly does not impact our conclusion that DFDs do not form sarkosyl-resistant smears that would be indicative of amyloid.

Expand on the concept of autoinhibited oligomerisation. Is this due to structural features? What might be the advantage of autoinhibited oligomerisation for these DFDs?

We have elaborated on this section in the results.

End of page 3, which "former set of adaptors" are referred to here? This is ambiguous.

We have replaced “former” with “innate immune”.

Page 5, the authors state that a kinetic barrier governs the activity of inflammatory signalosomes. While under the circumstances generated in this particular system, there is a kinetic barrier to the formation of large fibrillar complexes, can the same be said to be true in cells that respond to signals? They experience a specific triggering event. This should be redrafted to distinguish between the specific trigger in cells (downstream of a binding-driven event) and the kinetic barrier to self-association observed in this model system.

Yes, our findings establish that a kinetic barrier governs signalosome activation. By engineering a triggering event that is more specific than natural triggering events (see Figure 3), we exclude the possibility that the cell first responds to the signal to create conditions that stabilize inflammasome formation. This means that regardless of what may happen with a natural trigger, the driving force for assembly clearly pre-exists and is therefore held in check by a kinetic barrier.

On page 6, the statement "...lifespan may be limited by the thermodynamic drive for inflammatory signal amplification" is not clear. While this is strictly true following the initial triggering event, isn't lifespan limited by the stochastic activation? These very general statements stray beyond what can be substantiated on the basis of the data presented here.

We believe the source of confusion here was our misuse of the term “lifespan”. We have now replaced it with “life expectancy”, which we believe is substantiated by our statements as written.

Overall, the work presents a compelling, comprehensive analysis of the seeded self-assembly of DFDs. It identifies distinct properties for assembly of these domains that may underlie their particular physiological roles. However, some of the statements are quite general and not substantiated.Page 6. Is "end cell fate" the intended phrase?

We have revised the phrase.

The data regarding conservation of DFD-like modules and activity is interesting and probably deserves inclusion. However, without substantial evidence of expression levels (i.e., results) and a more complete understanding of these other systems, the statement "These results suggest that the function of DFDs as energy reservoirs preceded the evolution of animals" appears as an over-reach.

We demonstrated that sequence-encoded nucleation barriers of DFDs are shared across animal signalosomes (human, zebrafish, sponge). This is not trivial as such nucleation barriers are uncommon even among targeted screens of prion-like proteins.5 Therefore, they appear to have existed in the basal animal. We have now omitted the data concerning bacterial DFDs as these systems are indeed much less understood, and the concerned pathways lack the tripartite architecture of animal signalosomes. We therefore revised the sentence in question by replacing “evolution” with “radiation”.

Only a small number of DFDs exhibit this behaviour, so why is the conclusion drawn that energy storage for on-demand signalling may be the principal ancestral function of DFDs?

The totality of the data supports this conclusion. Briefly (but elaborated in the text), (1) intrinsic nucleation barriers are unusual even among self-associating proteins, the vast majority of which (e.g. condensates) would suffice for the only other major function ascribed to DFDs -- bringing effectors close enough for proximity-dependent activation (which has been repeatedly demonstrated in DFD-replacement experiments), (2) nucleation barriers are nevertheless conserved in innate immune signaling pathway, (3) that they are limited to approximately one DFD in each pathway is consistent with evolutionary selection to minimize accidental death.

Are there any other adapters like MyD88 that are inconsistent with this hypothesis? Are any others known to be controlled by oligomer formation? How strong is the evidence for hexameric oligomers? If there is a threshold size for oligomers, how does this differ from a stable seed/nucleus that triggers assembly, as in the discontinuous transition?

These are all good questions related to critiques that we have now addressed.

The use of the term "privatisation" is likely not consistently understood across the community and should be explained. Is it simply meant to imply independent operation? How is it actually different from other forms of deployment of DFDs that exhibit continuous assembly? Are they not also independent? What is implied by the opposite of privatisation here? The term may introduce ambiguity in this context.

We have now omitted this term.

Is there strong evidence that well-validated physiologically relevant LLPS systems exhibit supersaturation at concentrations that are very different from those of the DFDs examined in this study?

No, and this is a major point. As discussed in the text (with references), LLPS is incompatible with cell-wide supersaturation to a comparable magnitude as crystalline transitions, which precludes them from driving signal *amplification*. This helps to explain why the active state of DFD assemblies is ordered, when it has been repeatedly demonstrated that signal *propagation* itself does not require ordering.

The paragraph discussing TIR domains and functional amyloids would be enhanced with a comparison of amyloid systems where seeded nucleation results in assembly of a polymer with significant conformational change in the constituent monomers.

We do not yet understand how DFDs (and TIR domains) in some cases exhibit amyloid-like nucleation barriers without overt conformational differences between monomers and polymers. Work is underway in the lab to test specific hypotheses, but such discussion would be too speculative for the present paper.

The statement "High specificity also insulates pathways from each other" should be elaborated to discuss the issue of highly similar monomers that apparently assemble into filamentous forms with minimal structural rearrangement. How is the specificity generated?

We have elaborated the paragraph.

The final paragraph is speculative and utilises language that detracts from the quality and rigour of the study. While important principles have been revealed, more discussion of the limitations of the work would allow readers to evaluate the significance of the study and could be used to effectively stimulate further efforts to study the multiple different mechanisms that underpin critical signalling pathways in innate immunity and control cell fate.

We have now revised the final paragraph and included an extensive discussion of the limitations of the work.

**Reviewer #2 (Recommendations for the authors):**
(1) For clarity, it would be useful to include the names of the proteins in the bottom table of STable1, and such information at the top and bottom tables can be connected.

We are unable to determine what is meant by this suggestion. Table S1 does not have a “top” and “bottom table”. Every entry in Table S1 and S2 contains the protein name, its most frequently used alias in the literature (when not the official name), and the corresponding Uniprot protein ID.

(2) The language used in the abstract makes analogies between scientific and mundane terms, which compromises clarity. For example, what is meant by the terms shown below?(a) "......specifically templated by other DFDs....."

We have revised this phrase.

(b) "...function like batteries, storing and converting energy for life-or-death decisions."Batteries convert chemical energy into electrical energy or thermal energy. What is the electrical energy produced by DFDs? Is there any evidence that DFDs change the temperature of the cells or transfer heat?

We have now included a familiar example of a thermal battery that operates analogously to the manner we show for DFDs. As now elaborated extensively, such batteries operate via a physical rather than chemical process -- a change in the state of matter (solute to crystalline) of a supersaturated “phase change material” (this is an established term). This is exactly what we show is happening for DFDs. While it would be illustrative to measure the heat released upon DFD polymerization in cells, the much faster rate of heat transfer relative to molecular diffusion makes that impossible with present methods. Nevertheless, such measurements are unnecessary because disorder-to-order phase transitions are fundamentally exothermic.

(c) "....privatizing..."

We now avoid this term.

Using appropriate scientific terms to explain the scientific results presented in this manuscript will increase clarity. Analogously, it is difficult to understand what the title of the manuscript means, "Protein phase change batteries..."

We appreciate this critique and have removed “batteries” from the title to make the work more accessible to biologists. However, we reject the implication that such terminology is inappropriate. We presume the reviewer meant “unfamiliar” instead of “inappropriate”. The well-reasoned application of terms from other fields is standard practice and arguably essential to convey new concepts in biology. The modern biology lexicon is built on this. For example, Robert Hooke co-opted “cell” from the architecture of monasteries. More recently cell biologists appropriated “condensates” from soft matter physics. In both cases, the term while initially foreign to biologists usefully introduced a concept that lacked recognized precedent in biology. Similarly, “phase change battery” provides an accurate analogy for the central finding of our work, and we have now elaborated this analogy in the text.

Bibliography

(1) Garcia-Seisdedos, H., Empereur-Mot, C., Elad, N. & Levy, E. D. Proteins evolve on the edge of supramolecular self-assembly. Nature 548, 244–247 (2017).

(2) Alberti, S., Halfmann, R., King, O., Kapila, A. & Lindquist, S. A systematic survey identifies prions and illuminates sequence features of prionogenic proteins. Cell 137, 146–158 (2009).

(3) Kimbrough, H. et al. A tool to dissect heterotypic determinants of homotypic protein phase behavior. Protein Sci. 34, e70194 (2025).

(4) Glück, I. M. et al. Nanoscale organization of the endogenous ASC speck. iScience 26, 108382 (2023).

(5) Posey, A. E. et al. Mechanistic inferences from analysis of measurements of protein phase transitions in live cells. J. Mol. Biol. 433, 166848 (2021).